# Mid-Miocene warmth pushed fossil coral calcification to physiological limits in high-latitude reefs
Markus Reuter [1] ✉, Juan P. D'Olivo [2], Thomas C. Brachert[3], Philipp M. Spreter[3], Regina Mertz-Kraus [4] & Claudia Wrozyna [1]

The history of resilience of organisms over geologic timescales serves as a reference for predicting their response to future conditions. Here we use fossil *Porites* coral records of skeletal growth and environmental variability from the subtropical Central Paratethys Sea to assess coral resilience to past ocean warming and acidification. These records offer a unique perspective on the calcification performance and environmental tolerances of a major present-day reef builder during the globally warm mid-Miocene $CO_2$ maximum and subsequent climate transition (16 to 13 Ma). We found evidence for up-regulation of the pH and saturation state of the corals' calcifying fluid as a mechanism underlying past resilience. However, this physiological control on the internal carbonate chemistry was insufficient to counteract the sub-optimal environment, resulting in an extremely low calcification rate that likely affected reef framework accretion. Our findings emphasize the influence of latitudinal seasonality on the sensitivity of coral calcification to climate change.

The current focus on the consequences of anthropogenic $CO_2$ emissions for the marine biosphere has increased awareness of Earth's climatic history. This interest includes examining the response of tropical coral reefs, as the oceans' most biodiverse ecosystems, to past changes in seawater temperature and carbonate chemistry[1–5]. Assessing the combined effects of ocean warming and acidification on past reef crises is hampered by the lack of quantitatively verified, high-resolution proxy records, which document the environmental conditions and the physiological performance of reef-building organisms during such events[3]. The skeleton of massive tropical corals can serve as invaluable repositories of sub-annual to centennial information, encompassing a range of physical and chemical parameters intricately linked to environmental conditions[6]. Skeletal growth data is particularly informative as it serves both as an indicator of the former environment and the health of the organism[7]. However, since fossil corals are usually not preserved in their original composition, information on the skeletal growth parameters bulk density, linear extension rate and calcification rate and the calcification environment from the distant geological past is very limited[8]. Here, we use exceptionally well-preserved fossil skeletons of *Porites* corals as archives for environmental and calcification change in geologic time. *Porites* corals are major reef builders in the present day and are primarily used for coral growth studies and coral-based paleoclimate reconstructions in the Indo-Pacific region[6,7], which form the

basis for a quantitative assessment of the fossil proxy data obtained[3]. The corals of this study are notable for retaining their original aragonite mineralogy and porosity. This exceptional condition provides an unprecedented opportunity for geochemical proxy analysis and for recording the entire set of calcification parameters across the change from the global warmth of the Miocene Climatic Optimum (MCO, ~17 to 15 Ma) to the following cooling phase of the Middle Miocene Climate Transition (MMCT, until ~13 Ma) (Fig. 1a). The MCO is of particular interest as it exhibits characteristics expected in future climate change scenarios[9]. Intriguingly, coral reefs appear to have increased in abundance and volume worldwide during the MCO[10], despite above-modern global temperatures[11] and the lowest surface pH and aragonite saturation state ($\Omega_{ar}$) of the open ocean in the last 22 million years[12]. This raises the questions of whether the slow geological rate of change enabled buffering in the ocean[1,2] or adaption by reef builders[12], and whether ancient reef corals responded to high temperatures and ocean acidification in the same way as today's corals.

To examine how *Porites* biomineralization performed in warmer-than-current global climate during the most serious ocean acidification event of the late Cenozoic we used X-ray densitometry to measure skeletal bulk density, linear extension rate and calcification rate. Additionally, stable oxygen and carbon isotope ($\delta^{18}O$, $\delta^{13}C$) and elemental to calcium ratios (Sr/Ca, B/Ca) were determined to construct sub-annual proxy records for

[1]Institute of Geography and Geology, University of Greifswald, Greifswald, Germany. [2]Reef Systems Academic Unit, National Autonomous University of Mexico, Puerto Morelos, Mexico. [3]Institute for Earth System Science and Remote Sensing, Leipzig University, Leipzig, Germany. [4]Institute of Geosciences, Johannes Gutenberg University Mainz, Mainz, Germany. ✉e-mail: markus.reuter@uni-greifswald.de

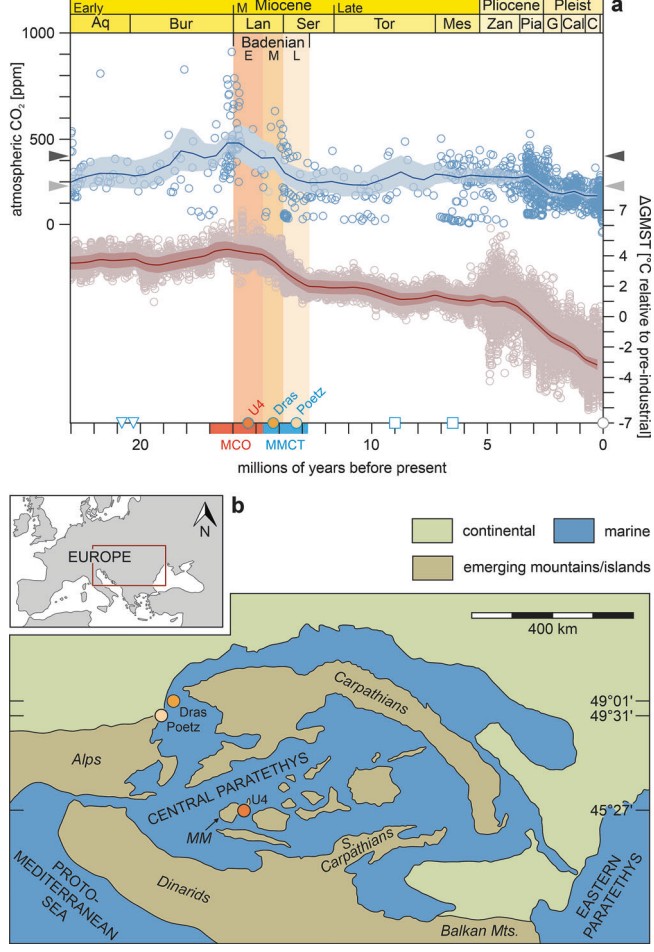

**Fig. 1 | Paleogeography of the Badenian Central Paratethys Sea and global climate evolution over the last 23.03 million years. a** Paleo-$CO_2$ (blue) and global mean surface temperature (GMST, red) records; proxy data (open circles) and statistically reconstructed 500-kyr mean values (solid lines) with 95% credible intervals[81]. The arrow heads indicate the atmospheric $CO_2$ levels in year 2023 (dark gray) and in pre-industrial times (light gray). The red and blue bars on the time-axis show the duration of the Miocene Climatic Optimum (MCO) and Middle Miocene Climate Transition (MMCT), respectively, and the colored vertical bars the temporal range of the early (E), middle (M) and late (L) Badenian regional stratigraphic substages. The symbols on the x-axis refer to Fig. 2 and indicate the temporal relationships of the coral growth records compared there. **b** Paleogeographic map[82] showing the location and paleolatitude of the sample sites (U4 = Hosszúhetény, Dras = Drasenhofen, Poetz = Pötzleinsdorf, MM = Mecsek Mts.). Paleolatitudes were inferred with GPlates (https://portal.gplates.org, last access 28-11-2023) using the Paleomap model by C. Scotese. The inset marks the position of the Central Paratethys within Europe.

temperature, endosymbiont photosynthetic activity, and internal carbonate chemistry. These insights into the biomineralization responses of a present-day reef-forming coral during a geological period of high $CO_2$ shed light on the evolutionary physiological adaptions and environmental limitations underlying the long-term patterns of coral reef change under climate change.

## Localities and stratigraphic background

The present study concentrates on three fossil *Porites* corals from the epicontinental Central Paratethys Sea in central and southeastern Europe (Fig. 1b). Stratigraphically, these corals represent the Badenian regional stratigraphic stage, corresponding to the Langhian and early Serravallian international stages (Fig. 1a). The Badenian is subdivided into three third-order stratigraphic cycles, which are referred to as the early Badenian

(15.97 to 14.7 Ma), the middle Badenian (14.7 to 13.8 Ma) and the late Badenian (13.8 to 12.7 Ma)[13] (Fig. 1a). The oldest coral used in this study, hereafter referred as U4, had a massive growth form (Supplementary Fig. 1a) and was collected near the village Hosszúhetény by the Mecsek Mountains (Hungary) (Fig. 1b). It represents the early Badenian[14]. The other two samples, a massive and a free-living (corallith) specimen named Poetz (Supplementary Fig. 1e) and Dras (Supplementary Fig. 1c), respectively, come from the Pötzleinsdorf and Drasenhofen localities in the Vienna Basin (Austria) (Fig. 1b). These corals represent the middle (Dras)[15] and late Badenian (Poetz)[16].

During the Badenian, the Central Paratethys is considered to have harbored the highest latitude warm-water reef system of the late Cenozoic (<23 Ma)[17]. This marginal reef system was characterized by a lack of extensive coral reefs and the dominance of low-diversity coral carpets and non-framework coral communities[18–20].

## Results and discussion

### Preservation, skeletal growth characteristics and geochemical patterns

Powder XRD analyses of the three fossil *Porites* skeletons revealed that they are composed of aragonite (<1 wt.% calcite). Microscopic inspection of the aragonitic skeletons yielded unaltered skeletal porosity (Supplementary Fig. 1b, d, f). X-ray positive images of all fossil coral skeletons (Supplementary Fig. 1a, c, e) show distinct alternations of thinner high-density bands (HDB, dark gray to black) and thicker low-density bands (LDB, light gray to white). These density variations, known from modern and fossil corals, reflect seasonal variations in linear extension and skeletal thickening (bulk density)[6–8,21,22]. A pair of HDB and LDB typically corresponds to one year of growth[6]. The product of linear extension rate (cm yr$^{-1}$) and bulk density (g cm$^{-3}$) is the calcification rate (g cm$^{-2}$ yr$^{-1}$)[7]. Of the samples analyzed, U4 has the lowest linear extension and calcification rates and the highest skeletal density, and Poetz has the highest linear extension and calcification rates, and the lowest skeletal density (Table 1). Dras shows comparatively intermediate values for all growth variables (Table 1). Overall, the calcification rates for the Middle Miocene *Porites* corals investigated in this study fall below the typical range of calcification rates for modern Indo-Pacific *Porites* (Fig. 2).

The δ$^{18}$O, Sr/Ca and B/Ca records of all fossil corals studied show regular cyclic variations, whereas the δ$^{13}$C exhibits a regular cyclic pattern only in the Poetz coral (Fig. 3). The systematic alignment of the geochemical proxy variations with the skeletal density banding patterns (Fig. 3) indicates a preservation of primary geochemical signals in all three *Porites* fossils and the annual nature of the recorded cycles. Nevertheless, trace element concentrations, particularly B, appear elevated in the fossil corals compared to modern tropical *Porites* (Supplementary Fig. 2). To evaluate the quality of measurement and calibration, we compare the JCp-1 values measured in this study with mean values for JCp-1 from the GeoReM database (http://georem.mpch-mainz.gwdg.de/, Application Version 27) based on literature data. Published values (Sr *n* = 8, B *n* = 6) range from 6670 ± 230 μg/g to 7500 ± ? μg/g for Sr and from 47.7 ± 1.2 μg/g to 52.4 ± 2.2 μg/g for B. Our JCp-1 values (Supplementary Table 1) are consistent with these ranges. Therefore, absolute values for Middle Miocene corals are inferred to reflect internal or external factors influencing trace element incorporation in the coral skeletons. When compared to modern corals, the fossil coral calcification and trace element values are closest to those from extreme and marginal *Porites* environments, such as the Galápagos, the Easter Island and the Ogasawara Islands (Fig. 2, Supplementary Fig. 2). Table 1 provides a summary of the variation in growth parameters and stable isotope and element to Ca ratios in and between the coral samples.

### Comparison with recent skeletal growth systematics in massive *Porites*

The calcification rate in annually-banded massive *Porites* from the Indo-Pacific is largely controlled by the linear extension (growth) rate, which negatively correlates with skeletal density[7] (Fig. 2). The explanation for this

relationship is that skeletal elements become thicker (denser) the longer they remain in the coral tissue layer, which moves upwards as the skeleton extends[23,24]. The skeletal growth parameters of the fossil corals in this study showed similar linear relationships and slopes as modern Indo-Pacific *Porites*, but lower y-intercepts (Fig. 2). The similar slopes indicate that the skeletal growth sensitivity of Middle Miocene *Porites* to environmental variations was equal to that of their modern counterparts and was not influenced by different coral morphologies (massive or corallith). However, the lower intercepts suggest that either baseline environmental conditions during the Middle Miocene were different than today or that calcification efficiency was reduced, e.g., due to different symbiont communities[25]. Diagenetic dissolution of skeletal aragonite can be excluded from influencing our results, since this only affects the density but not the linear extension rate and would therefore disturb the linear relationship between the two variables of skeletal growth.

## Indication for temperature-induced growth stress

Water temperature is the dominant factor controlling coral skeletal growth in well-lit shallow waters at the present day[7]. This control is evidenced in the positive relationship between site-averaged linear extension rates of massive *Porites* and the average annual sea surface temperature (SST)[26,27]. At the colony level, the relationship of coral growth and temperature is not linear. Increasing temperatures promote calcification until a taxon-specific thermal growth threshold is reached, after which the growth rate decreases[28,29].

For the three corals studied, $\delta^{18}O$ values and Sr/Ca ratios exhibit annual cycles in phase (Fig. 3), suggesting a common temperature control, where for both proxies, we assign low values to the warm season and high values to the cold season. No attempts were made to calculate absolute paleo-temperatures. This decision was due to the sensitivity of $\delta^{18}O$ values to hydrological effects[30], uncertainties regarding the spatio-temporal variability of the Sr/Ca composition of seawater in the past[31], and the large individual sensitivity of Sr/Ca ratios to SST observed in modern *Porites* corals[32,33]. However, the clear and symmetric $\delta^{18}O$ cycles in U4 and Poetz (Fig. 3a, c) indicate that the corresponding isotope signatures were largely controlled by seasonal temperature variations[34]. In contrast, the asymmetric, more irregular $\delta^{18}O$ cycles of Dras (Fig. 3b) suggest interference by other controlling factors (e.g., seasonal freshwater input, seasonal upwelling or colony rotation) in addition to seasonal SST changes driven by insolation[34]. Therefore, we use only the $\delta^{18}O$ values of U4 and Poetz, for estimating SST seasonality. Reconstructions of SST variability from $\delta^{18}O$ yield values of 10.5 ± 0.7 °C for U4 (years 1 to 5 and 11 to 13) and 9.9 ± 1.1 °C for Poetz (years 4 to 10) according to a mean coral calibration (−0.22 ‰ change in $\delta^{18}O$ per °C)[35] (Fig. 3a, c). To estimate a possible attenuation of the amplitude of annual $\delta^{18}O$ cycles associated with the very low (<0.6 cm yr$^{-1}$) linear extension rates[22], we correlated amplitudes and linear extension rates. No significant correlation was found for U4 and Poetz (Supplementary Fig. 3a, c). The temperature seasonality determined from $\delta^{18}O$ values appears therefore not significantly affected by growth rate effects in both corals. The reconstructed high annual SST range (Fig. 3a, c) is consistent with the high-latitude setting of the Central Paratethys reefs[17] and with the estimated SST seasonality of 9 to 10 °C for the MCO based on shell $\delta^{18}O$ of *Magallana gryphoides* oysters from the Vienna Basin[36]. The slope of $\Delta$Sr/Ca to $\Delta$SST is approximately –0.1 mmol/mol per °C for corals U4 and Poetz, based on the annual range in 5-point moving average Sr/Ca and SST range estimated from $\delta^{18}O$. This slope suggests roughly twice the temperature sensitivity observed in modern *Porites* corals (average –0.06[37]). However, in addition to temperature, Sr/Ca is also influenced by Rayleigh fractionation, driven by variations in the rate of aragonite precipitation and the rate of renewal of the calcifying fluid (cf)[32,38]. While these processes cannot be directly quantified here, large seasonal variability in carbonate ion concentration of the calcifying fluid ($[CO_3^{2-}]_{cf}$) can be inferred from the B/Ca data (see discussion below). Because the aragonite precipitation rate is considered proportional to the $[CO_3^{2-}]_{cf}$[39], seasonal differences in the rate of aragonite precipitation may contribute to the apparent high SST-Sr/Ca sensitivity observed in the fossil corals studied.

**Table 1 | Skeletal growth parameters and geochemical characteristics of Middle Miocene (Badenian) *Porites* corals from the Central Paratethys**

| Sample | Locality | Age [Ma] | Bulk density [g cm⁻³] | | | | | Linear extension rate [cm yr⁻¹] | | | | Calcification rate [g cm⁻² yr⁻¹] | | | | HDB timing | δ¹⁸O [‰] | | | | | δ¹³C [‰] | | | | B/Ca [µg/g] | | | | | Sr/Ca [µg/g] | | | |
|---|---|---|---|---|---|---|---|---|---|---|---|---|---|---|---|---|---|---|---|---|---|---|---|---|---|---|---|---|---|---|---|---|---|---|
| | | | yrs | Min. | Max. | Mean | SD | Min. | Max. | Mean | SD | Min. | Max. | Mean | SD | | n | Min. | Max. | Mean | SD | Min. | Max. | Mean | SD | n | Min. | Max. | Mean | SD | Min. | Max. | Mean | SD |
| U4 | Hosszúhetény, HU | 16.0–14.7 | 8 | 0.89 | 1.00 | 0.94 | 0.03 | 0.13 | 0.24 | 0.17 | 0.03 | 0.11 | 0.22 | 0.16 | 0.03 | summer | 182 | −4.61 | −1.86 | −3.41 | 0.77 | −3.02 | −0.04 | −1.54 | 0.55 | 222 | 41.08 | 80.60 | 59.38 | 8.94 | 7360 | 9467 | 8172 | 403 |
| Dras | Drasenhofen, AT | 14.7–13.8 | 8 | 0.75 | 0.93 | 0.84 | 0.06 | 0.22 | 0.38 | 0.30 | 0.05 | 0.20 | 0.31 | 0.25 | 0.05 | winter | 140 | −4.70 | −2.20 | −3.69 | 0.56 | −2.90 | 0.71 | −1.16 | 0.85 | 245 | 46.35 | 75.52 | 59.74 | 5.68 | 6975 | 8946 | 7905 | 302 |
| Poetz | Pötzleinsdorf (Vienna), AT | 13.8–12.7 | 9 | 0.65 | 0.83 | 0.74 | 0.06 | 0.39 | 0.69 | 0.49 | 0.06 | 0.32 | 0.45 | 0.36 | 0.10 | winter | 185 | −4.83 | −2.20 | −3.82 | 0.75 | −2.50 | 0.78 | −0.42 | 0.80 | 262 | 39.67 | 82.56 | 53.45 | 9.05 | 6314 | 7944 | 7093 | 266 |

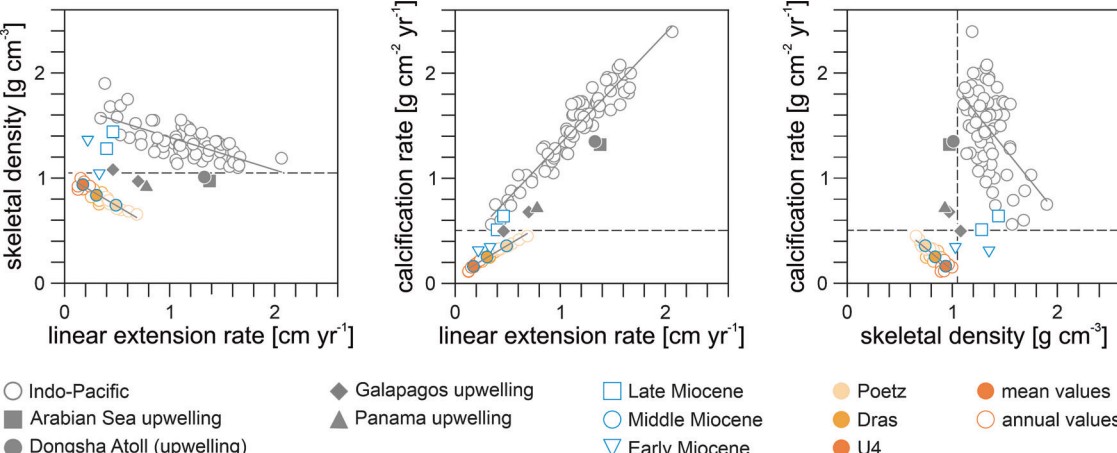

**Fig. 2 | Comparison of the skeletal growth variables of Miocene and recent massive *Porites*.** Recent growth records (mean values per site) are from various upwelling (filled gray symbols) and non-upwelling environments (open gray symbols) in the Indo- and Eastern Pacific. The blue symbols show fossil growth data (mean values per site) and the colored circles represent average annual values of the Middle Miocene corals in this study. The dashed lines show separation of Middle Miocene and recent (non-upwelling) corals into two groups. Fossil reference data (mean values per site) are from the eastern Atlantic (Early Miocene) and eastern Mediterranean Sea (Late Miocene). See Supplementary Data for locations and references of recent and fossil calcification data of massive *Porites*.

Although U4 and Poetz both show clear and symmetric annual $\delta^{18}$O cycles of similar amplitude, the cycles differ in shape and in their relationship to density. U4 exhibits narrow, sinusoidal cycles of $\delta^{18}$O due to slow growth in both winter and summer (Fig. 3a). This sinusoidal pattern is also evident in Sr/Ca and B/Ca (Fig. 3a). In U4, the formation of HDBs is associated with the warmest time of the year (Fig. 3a). This timing of HDBs could indicate a non-linear growth response beyond the optimal temperature[21] or redirection of resources from growth to reproduction in female colonies[40], a characteristic that cannot be determined in the studied samples. Another feature of U4 is an irregular fluctuating pattern in $\delta^{13}$C (Fig. 3a, Supplementary Fig. 4a, b). Notable is a large negative $\delta^{13}$C excursion in year 6, which coincides with an interval of unusually low linear extension rates, spanning from year 4 to year 9 (Fig. 3a, Supplementary Fig. 5). Temperature stress can destabilize coral endosymbiosis, thereby reducing photosynthetic input of $^{12}$C to the cf and thus decreasing skeletal $\delta^{13}$C values[41,42]. The high intra- and inter-annual fluctuation in $\delta^{13}$C values in U4 (Fig. 3a) could therefore be the expression of disturbed photosymbiosis under a sub-optimal (warm) temperature regime.

Once heat stress disrupts symbiosis, corals are deprived of their main sources of energy and carbon, which can lead to starvation and death. To maintain their metabolic needs, corals can increase their heterotrophic feeding[42]. This change in carbon source causes a decrease of $\delta^{13}$C values in the skeleton[41]. The coincidence of the negative $\delta^{13}$C excursion with the annual $\delta^{18}$O and Sr/Ca minima (Fig. 3a), i.e., the warmest time of the year, in U4 is consistent with a heat-induced breakdown of photosymbiosis, likely associated with bleaching (whitening of corals by the expulsion of the photosynthetic endosymbionts from the host tissue). Severe growth reductions over several years are common characteristics of symbiotic corals affected by thermal stress events[43–45]. The loss of $\delta^{18}$O cyclicity (Fig. 3a) and the strong positive correlation of $\delta^{18}$O with $\delta^{13}$C in the first three years after the assumed high temperature stress event (Supplementary Fig. 4c), when the coral grew most slowly according to the element/Ca chronologies (Fig. 3a), indicate kinetic isotope effects related to calcification rate[46]. This renders this part of the $\delta^{18}$O record of U4 unusable for reconstruction of SST seasonality. Conversely, Sr/Ca and B/Ca retained the corresponding seasonality, albeit with a reduced amplitude (Fig. 3a, Supplementary Fig. 5). The stress-induced decrease in extension rate could lead to an increase in bio-smoothing[47,48]. Bio-smoothing is described as the attenuation of the seasonal amplitude of the trace element records due to an overprinting of the older signature, while preserving the relationships between different trace elements[49]. This characteristic is evident in U4 (Fig. 3a). Further evidence for the influence of bio-smoothing on the seasonal trace element amplitude in

U4 comes from significant relationships between the seasonal amplitudes of Sr/Ca and B/Ca and the linear extension rate (Supplementary Fig. 3d, g). Furthermore, Sr/Ca shows a baseline shift towards more positive values after the stress event, but absolute B/Ca values remained unchanged (Fig. 3a, Supplementary Fig. 4f). This baseline shift in Sr/Ca cannot be attributed to inter-annual temperature variability as it is not mirrored in $\delta^{18}$O (Fig. 3a). Interestingly, the opposite dynamic was also observed after coral bleaching, i.e., a long-term change in $\delta^{18}$O but no change in Sr/Ca[50]. However, consistent with our observations, an extended disruption in several trace element ratios was found following a bleaching event, leading to changes in seasonality and baseline shifts that differently affected individual trace elements[49]. This heterogeneous response likely reflects a variety of physiological alterations acting on the different mechanisms by which individual elements are incorporated into the coral skeleton[43,49].

The $\delta^{18}$O, Sr/Ca and B/Ca annual cycles of Poetz are characterized by broad, rounded minima during the warm season, and narrow, acute peaks during the cold season (Fig. 3c). This arc-shaped or cuspate pattern[34] reflects a higher seasonal difference in growth rate compared to U4, with high summer extension rate and a significant decrease, or possibly cessation, of skeletal extension in winter[51,52], resulting in HDB formation. Additionally, the $\delta^{13}$C signal shows a distinct seasonality which is inversely correlated to $\delta^{18}$O (Fig. 3c, Supplementary Fig. 4e). The seasonal $\delta^{13}$C pattern in Poetz likely reflects light- and temperature-induced changes in the metabolic supply of dissolved inorganic carbon (DIC) provided by the photosynthetic endosymbionts[41]. The regular B/Ca fluctuations shown in Fig. 3c are consistent with seasonal variations in the abundance of symbiont-derived DIC[38]. Modern *Porites* corals adjust the pH of their cf in response to seasonal variations of the DIC pool available for calcification. This adjustment leads to large antithetic seasonal fluctuations in the pH and $[CO_3^{2-}]_{cf}$[38]. Boron incorporation into the coral aragonite depends primarily on $[CO_3^{2-}]_{cf}$[53,54], and thus the B/Ca ratio of the coral skeleton is sensitive to both environmental influences modulating $[CO_3^{2-}]_{cf}$ and biological regulation of the internal carbonate system[38,43]. The large seasonal B/Ca variations, with maximum in winter and minimum in summer, which occur in all fossil corals studied (Fig. 3), are therefore consistent with a biological control on calcifying fluid $\Omega_{ar}$ against external driven variations in the $[CO_3^{2-}]_{cf}$ similar to that observed in modern corals[38]. Interestingly, B/Ca values in U4 remain stable before and after the presumed bleaching stress event (Fig. 3a). Although boron isotope data are not available to fully reconstruct $[CO_3^{2-}]_{cf}$ or $pH_{cf}$, this observation strongly indicates that U4 maintained the minimum $\Omega_{cf}$ required for calcification, albeit at reduced rates. Dras

**Fig. 3 | Seasonal coral proxy records. a** Early Langhian/early Badenian (U4). **b** Late Langhian/middle Badenian (Dras). **c** Early Serravallian/late Badenian (Poetz). Thin gray lines in the density and element/Ca records depict measured value resolution, thick black lines 5-point running averages. Circle symbols in the stable isotope records indicate measured data points. Blue vertical lines define years. Shaded area in (**a**) highlights the growth anomaly detected in coral U4 (Supplementary Fig. 2). The vertical red line in (**a**) indicates the timing of the inferred bleaching event and the horizontal green line in the corresponding Sr/Ca record mean values for the data prior to and after the stress event. The vertical red line in (**b**) marks a growth hiatus that is recognizable in the X-ray image (Supplementary Fig. 1c). Note the inverted y-axes for density, δ¹⁸O and Sr/Ca.

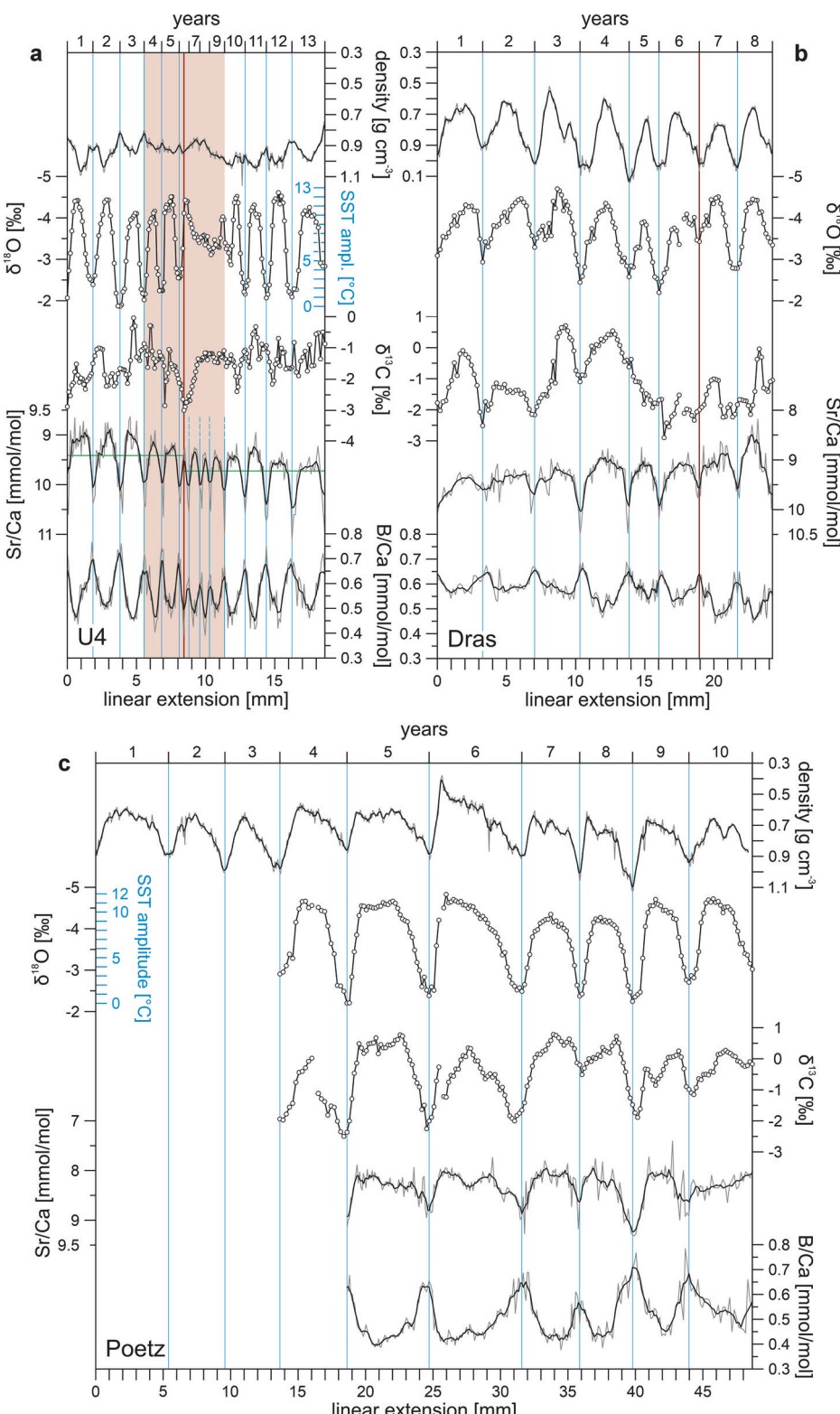

shows the same cuspate Sr/Ca and B/Ca patterns and timing of HDBs as Poetz (Fig. 3b, c), indicating reduced growth due to low winter insolation.

From the low extension rates observed in U4 during summer and winter, the timing of HDBs in summer, and geochemical evidence of summer physiological stress, we infer that warm season SSTs in the Central Paratethys surpassed the optimal range for *Porites* growth during the MCO. Meanwhile, winter SSTs approached the lower limit for calcification. This interpretation is consistent with the high δ¹⁸O-SST seasonal temperature

difference of 10.5 ± 0.7 °C in U4 and with estimates of seasonal SSTs that derived from an actualistic comparision of the Central Paratethyan echinoderm fauna with the nearest living relatives and their climatic tolerances[55]. The echinoderm-based SST reconstruction indicates a seasonal variation from 17 to 18 °C in winter to at least 28 °C in summer in the southern and central parts of the Central Paratethys during the early Badenian[55]. It is important to note that thermal thresholds vary between reef regions[56]. High-latitude corals can be sensitive to elevated temperatures that are well within

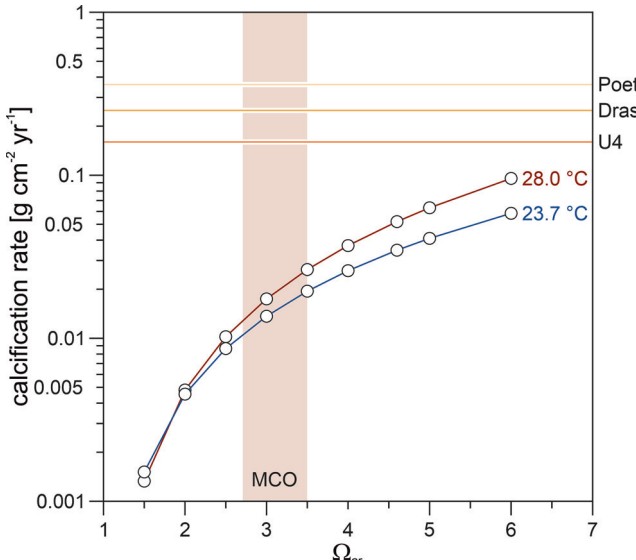

**Fig. 4 | Inorganic aragonite precipitation rates as a function of seawater $\Omega_{ar}$ at 23.7 °C (estimated mean annual SST for Poetz based on linear extension) and 28 °C (estimated early Badenian summer SST in the southern and central parts of the Central Paratethys[55]).** The red vertical bar marks the reconstructed range of surface ocean $\Omega_{ar}$ of 2.7 to 3.5 for the MCO[12]. Average calcification rates of the three fossil *Porites* corals studied (colored horizontal lines) are far from inorganic precipitation, consistent with carbonate chemistry up-regulation of the calcifying fluid[80].

the normal range for lower-latitude corals[57]. The cuspate shape of the annual cycles and the shift in HDB formation to winter in the corals from the MMCT (Dras, Poetz) indicate growth conditions improved during summer and deteriorated in winter. Consistently, the echinoderm evidence indicates a decrease of late Badenian winter SSTs to about 15 to 16 °C[55]. The deviating seasonal patterns in U4 (sinusoidal) and Dras/Poetz (cuspate) indicate a period of climatic cooling, consistent with the global climate trend[11] (Fig. 1a) and the retreat of mangroves from the Central Paratethys following the MCO[58].

## Effect of ocean acidity

The skeletal density of the studied corals is only half (U4 57 ± 2%, Dras 52 ± 3% and Poetz 48 ± 3%) of the density expected from the modern extension rate–density relationship for massive *Porites* in the Indo-Pacific (Fig. 2) and must therefore have been influenced by a factor not affecting linear extension rate. Modeling of *Porites* skeletal growth as a function of seawater carbonate chemistry has shown that density growth is sensitive to changes in $[CO_3^{2-}]_{sw}$, whereas extension growth is not[24]. The same response in coral skeletal density has been observed at naturally low-pH sites[59,60] and as a long-term effect of ocean acidification[61,62]. Therefore, the low density values of the Middle Miocene corals could be explained by low $[CO_3^{2-}]_{sw}$. We use published skeletal density data of late Cenozoic reef corals to estimate the relative magnitude of ocean acidification during the Middle Miocene. Although, the available data is limited, the existing records, ranging from the Early Miocene to the early Pleistocene, consistently show lower calcification performance compared to the present (hypo-calcification), regardless of taxon, stratigraphic age, ocean basin and paleo-latitude[8]. Hypo-calcification of corals may therefore have been the normal state in the late Cenozoic, probably due to a different baseline carbonate chemistry of seawater than today[8].

To assess the deviation from the Miocene baseline and reduce taxonomic bias[63], the mean annual density of the studied *Porites* fossils is compared with massive *Porites* data from the Early and Late Miocene as well as from present-day low-pH upwelling environments (Fig. 2). This comparison reveals a lower skeletal density for the Middle Miocene than for the

Early and Late Miocene. Middle Miocene density values are in the range (U4) or lower (Dras, Poetz) than those observed in *Porites* from present-day upwelling zones (Fig. 2). In contrast, the skeletal density of Early and Late Miocene *Porites* largely overlaps with that of modern *Porites* from non-upwelling sites in the Indo-Pacific (Fig. 2). Although the Early and Late Miocene *Porites* also have low calcification rates, these are a product of their low extension rates (Fig. 2) and thus do not indicate a low seawater pH or $\Omega_{ar}$. Interestingly, all Miocene *Porites* calcification data originate from the northern margin of coral reef distribution[20]. The very low extension rates of these corals may therefore reflect the broader latitudinal extent of the Miocene tropical reef belt compared to today[17]. Also interestingly, Early Miocene calcification rates are approaching the mid-Miocene trend, while Late Miocene rates are at the low-end of the modern trend (Fig. 2), a pattern that seems consistent with the general Neogene cooling (Fig. 1a). The lower skeletal density of the *Porites* studied in relation to the Early and Late Miocene *Porites* (Fig. 2) provides indirect evidence for aragonite undersaturated/low pH conditions in the Central Paratethys during the early Middle Miocene, in accordance with the open ocean trends of surface water pH and $\Omega_{ar}$ in the late Cenozoic[12]. However, the apparent lack of reduction in coral skeletal density during the MCO in the data from the Central Paratethys (Fig. 2) seems to indicate that acidification had no negative effects at the highest atmospheric $CO_2$ concentrations (Fig. 1a). This divergence between the trends of coral skeletal density and global $CO_2$ could stem from changes in local $[CO_3^{2-}]_{sw}$, which may have differed from the changes in overall ocean $[CO_3^{2-}]$, both in absolute values and seasonality, due to the restricted nature of the Central Paratethys basin (Fig. 1b). In contrast to Sr/Ca, B/Ca appears to be unaffected by Rayleigh fractionation and determined solely by the carbonate chemistry of the cf[38,54]. Therefore, the elevated B/Ca ratios and inferred low $[CO_3^{2-}]_{cf}$ of Middle Miocene *Porites* compared to modern *Porites* and of U4 and Dras compared to Poetz (Supplementary Fig. 2) are consistent with reduced $[CO_3^{2-}]_{sw}$ during the MCO. However, clearly determining external influences on the B/Ca composition of Miocene coral skeletons is complicated by uncertainties in the internal pH-upregulation and in seawater elemental concentrations, carbonate chemistry, and nutrient levels on both global and local scales. In the data generated in this study, acidification-induced changes in coral skeletal density were probably masked by sub-optimal (cold and warm) temperature exposure, resulting in a low linear extension rate. The slow linear extension promoted thickening of the coral skeleton by increasing the duration each skeletal element remained in contact with the coral tissue layer[24]. This skeletal thickening would have compensated to some extent for the effects of ocean acidification on the annual bulk density of U4.

Corals in the Central Paratethys likely exerted strong control over their carbonate chemistry to maintain cf homeostasis under persistently low seawater pH and saturation state. This control is similar to that of modern massive *Porites* growing at $CO_2$ seeps in Papua New Guinea, where the measured $pH_{sw}$ of 7.4 ($DIC_{sw}$ 2235 µmol kg$^{-1}$)[64] is even lower than the estimated MCO value of 7.6 ± 0.1[12]. The ability of $pH_{cf}$ up-regulation is inferred from the clear seasonal B/Ca in the Middle Miocene *Porites* studied (Fig. 3), consistent with observation in present-day corals, in which seasonal cycles in B/Ca are associated with metabolic DIC supply[38]. Despite their calcification rates being lower than those of their modern counterparts, the studied fossil corals still greatly exceeded inorganic aragonite precipitation rates (Fig. 4). From these elevated rates it can be assumed that the physiological mechanisms to promote calcification and mitigate ocean acidification were already active during the MCO. Nevertheless, $CO_2$ exposure appears to affect a variety of physiological processes integral to host-symbiont dynamics and photophysiology[65], potentially influencing bleaching thresholds[66] and post-bleaching recovery[65]. The extremely low calcification rate of U4 due to a low skeletal density and linear extension (Fig. 2), along with the sensitivity to external pH and seawater carbonate chemistry variations, indicate that the interaction of high surface ocean $CO_2$ and strong latitudinal temperature seasonality brought even tolerant coral taxa in the high-latitude reef ecosystem of the Central Paratethys to their adaptive limits during the MCO. In contrast, the similar density values of

Early Miocene, Late Miocene and modern *Porites* (Fig. 2) imply that the lower levels of ocean acidification during the Early and Late Miocene[12] could be balanced by carbonate chemistry up-regulation of the corals' cf.

*Porites* corals were important reef builders in the Central Paratethys[18,19]. The extremely low calcification rates of *Porites* therefore likely compromised reef net carbonate production and accretion rates in this region[67]. Additionally, the low bulk density of the corals reduced the mechanical strength of their skeletons[59], and their slow linear extension limited their ability to compete for space[68]. This leads us to infer that the low calcification performance of *Porites* hampered the formation and maintenance of complex, three-dimensional coral reef frameworks in the Central Paratethys. We conclude that globally-warm, high-$CO_2$ conditions are likely to be more limiting to coral calcification and reef build-up in seasonal non-tropical environments than in the more stable tropical regions, diminishing the potential for subtropical zones to act as climate refuge for the threatened biodiversity of tropical coral reefs.

## Methods

### Sample origin and deposition
The Dras and Poetz corals come from the collection of the Natural History Museum Vienna (Austria) (NHMW 2024/0194/0001 and NHMW 2024/0194/0002) and were collected at the beginning of the last century from historical outcrops that are now inaccessible. Sample U4 is deposited in the geological and paleontological collection of Greifswald University (Germany) (GG 520).

### Taxonomic remarks
The taxonomy of *Porites* is one of the most challenging among the Scleractinia. This is due to the high variability of skeletal structures within individual coralla and geographic variation across the extensive range of many nominal species. Furthermore, the incongruence between morphological and molecular systematics has left the delimitation of many species uncertain[69,70]. For these reasons, *Porites*-based coral (paleo-)climate and calcification studies avoid species-level differentiation, which also applies to the reference dataset of modern and fossil *Porites* used in this study (Fig. 2). Given the minimal differences among the Miocene *Porites* species in the Mediterranean region[71], we have therefore also refrained from determining the fossil *Porites* presented here to the species level.

### Sample preparation
The *Porites* fossils were sectioned parallel to the growth axis into slices that were ground to planar slabs of 6 mm thickness. The slabs were cleaned in an ultrasonic bath in de-ionized water and dried at 38 °C. Geochemical analyses were performed on a 4 mm wide rod, which was cut from each coral slab following the density transect and subsequently cleaned and dried in the same way as described for the slabs. Prior to oxygen and carbon stable isotope sampling, the sample rods, first used for the trace element analysis, were impregnated with composite dental resin (Estelite Universal Flow) as a stabilizing medium and to prevent contamination through infiltration of the milled powders into the pore spaces[72].

### Diagenetic screening
To check whether diagenetic calcite was present, powder generated during planar grinding of the slabs, and thus averaging over their entire surface, was used for powder X-ray diffraction (XRD) analysis at the Institute for Earth System Science and Remote Sensing, Leipzig University (Germany) with a Rigaku Miniflex diffractometer with scanning angles of 20° to 60° 2θ. The detection limit of the method is ~1 wt.%. The coral slabs were X-rayed using a digital X-ray cabinet (SHR 50 V) at the Leipzig University, to reveal the high- and low-density band patterns, possible internal diagenetic alterations, and biogenic encrustations and borings. Additionally, the surface of the coral slabs was imaged with a Keyence VHX-7000 digital microscope to examine the impact of dissolution and cementation on skeletal porosity. The fossil *Porites* corals from the Early and Late Miocene, used to estimating the relative magnitude of ocean acidification during the Middle Miocene

(Fig. 2), were also examined for diagenetic changes and state of preservation, and found to be comparable to the Middle Miocene corals of this study.

### X-ray densitometry
Calcification parameters were measured for all fossil *Porites* fossils analyzed in this study (Early Miocene, Middle Miocene, Late Miocene) according to the same protocol and laboratory standards[8]. For quantification of the skeletal bulk density, we used the X-ray densitometry app Coral*XDS* (https://hcas.nova.edu/tools-and-resources/coralxds/). Density was measured at a 0.05 mm resolution on the digital X-ray images of the coral slabs along 3 mm wide transects parallel to the corallites along the main growth. Breakouts in the skeleton, biogenic encrustations and borings were avoided. Gray scale-density calibrations of the X-ray images were performed by using measuring standards for zero density (air, $\rho = 0$ g cm$^{-3}$) and for solid aragonite (*Tridacna* shell, $\rho = 2.93$ g cm$^{-3}$) with the same thickness as the coral slabs. Linear extension rate and bulk density for the corresponding year were derived from the cool season density peaks of adjacent annual density bands. The calcification rate was calculated based on Eq. (1):

$$\text{calcification rate}\left(\text{g cm}^{-2}\text{yr}^{-1}\right) = \text{linear extension rate}\left(\text{cm yr}^{-1}\right) \\ \times \text{density}\left(\text{g cm}^{-3}\right) \tag{1}$$

### Trace element analysis
Trace element concentrations were determined at the Institute of Geosciences, Johannes Gutenberg University Mainz (Germany), using an Agilent 7500ce inductively coupled plasma-mass spectrometer (ICP-MS) coupled to an ESI NWR193 ArF excimer laser ablation (LA) system equipped with a TwoVol2 ablation cell. The ArF LA system was operated at a pulse repetition rate of 10 Hz and an energy density of ca. 3.5 J cm$^{-2}$. Ablation was carried out under a He atmosphere and the He-sample aerosol was mixed with Ar before entering the plasma. Measurement spots with a beam diameter of 80 μm were aligned with a midpoint distance of 100 μm along transects following individual vertical skeletal elements. Backgrounds were measured for 15 s prior to each ablation. Ablation time was 30 s, followed by 20 s of wash out. Of the monitored isotopes, $^{11}$B, $^{43}$Ca and $^{88}$Sr were used for this study. Signals were recorded in time-resolved mode and processed using an in-house Excel spreadsheet[73]. Details of the calculations are given in Mischel et al.[74]. NIST SRM 610 and 612 were used as calibration material, applying the reference values reported in the GeoReM database[75,76] to calculate the element concentrations of the sample measurements. During each session, basaltic USGS BCR-2G, synthetic carbonate USGS MACS-3 and biogenic carbonate (JCp-1-NP) were analyzed repeatedly as quality control materials to monitor precision and accuracy of the measurements as well as calibration strategy. All reference materials were analyzed at the beginning and at the end of a sequence and after ca. 40 spots on the samples. For all materials, $^{43}$Ca was used as internal standard applying for USGS BCR-2G and MACS-3 the preferred values reported in the GeoReM database, for JCp-1-NP a Ca content of 38.18 wt.%[77] and 39 wt.% for the samples[78]. Resulting element concentrations for the quality control materials together with reference values are provided in Supplementary Table 1. Element concentrations for the samples are converted into molar ratios of Ca, i.e., B/Ca and Sr/Ca.

### Stable oxygen and carbon analysis
For oxygen ($\delta^{18}$O) and carbon ($\delta^{13}$C) stable isotope sampling, we choose a micro-sampling milling approach[72], suitable to produce seasonal records from corals with small extension rates[22]. Briefly, the coral sample rods were mounted on a manually operated XYZ table of a micro-milling system and milled down at regular increments[72]. The width of sampling increments was 0.1 mm for specimen Poetz and 0.05 mm for specimens Dras and U4, to ensure a sub-monthly resolution. Every second powder sample from Poetz and every third sample from Dras U4 were analyzed. To capture the full range in the $\delta^{18}$O seasonality as best as possible, samples adjacent to a $\delta^{18}$O seasonal peak were also analyzed. The isotopic analyses were conducted at

the Institute for Earth System Science and Remote Sensing, Leipzig University (Germany). Carbonate powders were reacted with 105% phosphoric acid at 70 °C using a Kiel IV online carbonate preparation line connected to a MAT 253 isotope ratio mass spectrometer. All carbonate values are reported in per mil (‰) relative to the VPDB standard. Reproducibility was monitored by replicate analysis of laboratory standards (8 standards per tablet of 46 samples) and was better than ±0.06‰ (1σ) for $\delta^{13}C$ and better than ±0.08‰ (1σ) for $\delta^{18}O$.

## Age models

Densitometric, trace element and stable isotope analyses were carried out on the same piece of the coral skeleton (rod) at regular intervals, which facilitates direct comparisons between the different proxies. However, differences in sampling methodologies can lead to slight misalignments. This was avoided by matching the cool-season peaks of $\delta^{18}O$ and Sr/Ca to the density record. Density was used as the reference record as it defines the growth rate. The cool-season peaks were used as tie points because they are more narrowly defined as the warm-season peaks.

## Calculation of inorganic aragonite precipitation rates

Inorganic aragonite precipitation (Fig. 4) was calculated based on the abiotic kinetic parameters for aragonite precipitation in seawater as a function of temperature[79], using a similar approach to McCulloch et al.[80]. Briefly, values were estimated applying Eq. (2):

$$R = k(\Omega - 1)^n \tag{2}$$

where $R$ = precipitation rate (μmol m$^2$ h$^{-1}$), $k$ = rate constant (μmol m$^2$ h$^{-1}$), $\Omega$ = saturation state, and $n$ = order of reaction. Both $k$ and $n$ are a function of temperature according to Eq. (3) and Eq. (4):

$$k = -0.01777 \times T^2 + 1.47 \times T + 14.9 \tag{3}$$

$$n = 0.0628 \times T + 0.0985 \tag{4}$$

where $T$ is the temperature in degree Celsius (°C). We used a lower temperature estimate of 23.7 °C as inferred for Poetz using the published linear regression equation linking extension rate of modern massive *Porites* in the Indo-Pacific region to annual SST[7]. This equation cannot be applied to determine the annual SST for U4 because it does not consider the non-linear growth above the upper thermal stress threshold. We therefore use an upper temperature estimate of 28 °C based on the summer SST estimate for the southern and central parts of the Central Paratethys during the early Badenian[55]. Precipitation rate in μmol m$^2$ h$^{-1}$, was converted to g m$^2$ yr$^{-1}$ using a molecular weight for CaCO$_3$ of 100.09 g per mole.

## Reporting summary

Further information on research design is available in the Nature Portfolio Reporting Summary linked to this article.

## Data availability

All data generated for this study are included in this published article and its Supplementary Data file.

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

## Acknowledgements

This research was supported by funding provided from the Austrian Science Fund (FWF, grant P 29158-N29 awarded to M.R.). We thank Nàdai László (Hungary) for providing the sample U4. We would also thank Mathias Harzhauser (Natural History Museum Vienna, Austria) and Thomas A. Neubauer (Bavarian State Collection for Palaeontology and Geology, Germany) for discussions on Central Paratethys stratigraphy and paleogeography.

## Author contributions

M.R. conceived and designed the study, wrote the manuscript and created the figures with support from J.P.D. and P.M.S. performed the X-ray densitometry, stable isotope and trace element analyses under the supervision of T.C.B. and R.M.-K. C.W. contributed to the statistics and data analysis. M.R., J.P.D., T.C.B., R.M.-K., and C.W. contributed to the data interpretation and discussion.

## Funding

## Competing interests

The authors declare no competing interests.
