## [Transparent Peer Review file · Communications Earth & Environment]

Mid-Miocene warmth pushed fossil coral calcification to physiological limits in high-latitude reefs

Corresponding Author: Dr Markus Reuter

Version 0:

Decision Letter:

Dear Dr Reuter,

Your manuscript titled "Coral calcification at its limits during mid-Miocene warmth" has now been seen by 2 reviewers, and we include their comments at the end of this message. They find your work of interest, but raised some points that need addressing, such as related to the data presentation in the figures and text and clarification of uncertainties associated to some of the data.

We are interested in the possibility of publishing your study in Communications Earth & Environment, but would like to consider your responses to these concerns and assess a revised manuscript before we make a final decision on publication.

We therefore invite you to revise and resubmit your manuscript, along with a point-by-point response that takes into account the points raised. Please highlight all changes in the manuscript text file.

Please submit your point-by-point responses as a separate file, distinct from your cover letter where you can add responses to the Editors' comments that you do not want to be made available to the reviewers. Word files are preferred. We recommend that any figures, tables or graphs that are included in the response to reviewers are also included in the main article or Supplementary Information.

Please use the following link to submit your revised manuscript, point-by-point response to the referees' comments (which should be in a separate document to any cover letter), a tracked-changes version of the manuscript (as a PDF file) and the completed checklist:

Link Redacted

We hope to receive your revised paper within six weeks; please let us know if you aren't able to submit it within this time so that we can discuss how best to proceed. If we don't hear from you, and the revision process takes significantly longer, we may close your file. In this event, we will still be happy to reconsider your paper at a later date, as long as nothing similar has been accepted for publication at Communications Earth & Environment or published elsewhere in the meantime.

Please do not hesitate to contact us if you have any questions or would like to discuss these revisions further. We look forward to seeing the revised manuscript and thank you for the opportunity to review your work.

Best regards,

Nadine Schubert, PhD
Editorial Board Member
Communications Earth & Environment
orcid.org/0000-0001-7161-7882

EDITORIAL POLICIES AND FORMATTING

Editorial Policy: [Policy requirements](https://www.nature.com/documents/nr-editorial-policy-checklist.pdf) (Download the link to your computer as a PDF.)

- Behavioural and social science
- Ecological, evolutionary & environmental sciences
- Life sciences

<https://www.nature.com/documents/nr-reporting-summary.zip>

Furthermore, please align your manuscript with our format requirements, which are summarized on the following checklist: [Communications Earth & Environment formatting checklist](https://www.nature.com/documents/commsj-phys-style-formatting-checklist-article.pdf)

and also in our style and formatting guide [Communications Earth & Environment formatting guide](https://www.nature.com/documents/commsj-phys-style-formatting-guide-accept.pdf) .

*** DATA: Communications Earth & Environment endorses the principles of the Enabling FAIR data project (<http://www.copdess.org/enabling-fair-data-project/>). We ask authors to make the data that support their conclusions available in permanent, publically accessible data repositories. (Please contact the editor if you are unable to make your data available).

All Communications Earth & Environment manuscripts must include a section titled "Data Availability" at the end of the Methods section or main text (if no Methods). More information on this policy, is available at <http://www.nature.com/authors/policies/data/data-availability-statements-data-citations.pdf>.

If a community resource is unavailable, data can be submitted to generalist repositories such as [figshare](https://figshare.com/) or [Dryad Digital Repository](http://datadryad.org/). Please provide a unique identifier for the data (for example a DOI or a permanent URL) in the data availability statement, if possible. If the repository does not provide identifiers, we encourage authors to supply the search terms that will return the data. For data that have been obtained from publically available sources, please provide a URL and the specific data product name in the data availability statement. Data with a DOI should be further cited in the methods reference section.

REVIEWER COMMENTS:

Reviewer #2 (Remarks to the Author):

I was asked to review "Coral calcification at its limits during mid-Miocene warmth." I find that the paper is of interest to a general geologic audience and that the findings are important. However, this journal is designed for a general Earth science audience, and the way the article is laid out right now makes it hard to read. Important facts are getting lost in the article (I was not aware until the end that these were subtropical corals). Also, the different coral sites are getting mixed up. I will have more specific comments below, but I highly recommend a results section where you detail what the cores showed regarding growth rates and geochemical measurements before a separate discussion section. This would make it easier to follow your

reasoning.

Abstract: You need to mention that these corals are growing in subtropical conditions here. In your last sentence, you state that the important finding is that corals were stressed in subtropical environments.

Line 43-44: "However, the preservation of all growth parameters (bulk density, linear extension rate and calcification rate) and of the original geochemical composition is a very rare feature in fossil corals, particularly further back in the geological record." This sentence reads very awkwardly because it runs on a little. I would look over the intro paragraph again because some of the sentences could be broken up and read better.

Line 52: It is becoming more common to use Miocene Climatic Optimum (MCO) instead of the MMCT because it is now thought that the MCO extended into the Early Miocene.

Line 54: It showed is not correct surely it is it shows in the present tense.

Line 74-87: Shouldn't this be in the intro or methods? I would not categorize this as a result.

Figure 1: make the site locations brighter. I had trouble finding site 1 on the figure also, you might consider adding the nicknames (U4 etc.) to the site map and age figure above.

General Comment: I would find it helpful if you included a table here showing the age location and maybe some major measurements for the different corals. I got lost in the article, trying to remember which coral corresponded to which period. Doing this would make it easier to find this information.

Lines 100-105: This would be much better in the table mentioned above. Right now, it is jumping around a lot, and that makes it hard to read. It would also save you some space.

Line 144: It should be the interpretation

Line 156: as stated above this needs to be said earlier

Lines 158-216: I would consider switching the discussion of U4 before Poetz. It is a little odd to talk about the older coral second and then the MMCT before the MCO. At the very least, I would mention in both these paragraphs the age and whether these are MMCT or MCO corals because this would make it easier to follow the importance of the seasonal signals.

170-174: this discussion of B/Ca seems to be in the wrong place it really should be a separate paragraph before or after the discussion of the coral signals.

Lines 192-194: This seems to be a really important finding, given that there are very few bleaching records during the Miocene. Right now, it is buried in the center of this VERY long paragraph. Maybe a paragraph break here would emphasize this finding better.

Lines 219-220: Where do these SSTs come from? if they are assumed from the low summer growth rates above this has to be stated. However, given that the authors earlier stated that they would not reconstruct SSTs from $\delta^{18}\text{O}$ (correctly) then there is no concrete evidence that SSTs were above optimal summer growth rates. Therefore, this finding needs to be explained better or qualified.

Lines 222-225: See above. It is also worth noting that other processes can cause stress and bleaching. These statements need to be explained better.

Lines 284-287: This needs to be stated in the abstract as this is the first mention of subtropical conditions.

Overall this is a significant study but needs some work to make the results more accessible to the general geologic community. I am convinced that this is possible with Minor Corrections.

Reviewer #3 (Remarks to the Author):

Review of "Coral calcification at its limits during mid-Miocene warmth."

Reuter et al. use Scleractinian coral data from the Middle Miocene to show that corals neither thrived, nor died out significantly during an extensive period of elevated CO_2 and higher than present temperatures. The manuscript is well written, and of broad interest to the palaeoclimate and coral science community. I have a few comments which I hope will

improve the manuscript before its eventual publication.

The geochemical data which is key to this study is not presented as the best quality as given here. Tucked away in table S1 are some differences of 14–42% (in B) away from the carbonate standards, which is quite extreme. And the uncertainty is presented in a different unit to that plotted in the main figures ($\mu\text{g/g}$ vs. mmol/mol). It seems the lab has measured high here, is that corrected for in the data (which also reads high compared to most published modern data)? Or would you propose the published values are too low?

Being more upfront about the uncertainty and data values would be helpful to understanding the data in more detail. E.g. does a boron/calcium of $700\mu\text{mol/mol}$ really imply a low CO_2 - or could it be something else? (e.g. Standish et al. 2023, GCA).

It is a shame not to see the relative temperature reconstructions made more of, the authors make a great deal out of the exceptional preservation and original aragonite of the corals, and then consider the seasonality, but do not show it anywhere.

The above links into a point made on L 163 where the authors talk about temperatures low enough to be unfavourable to calcification, does this imply a much greater seasonality than the modern? Only a few locations now have tropical corals which completely stop extension during cold periods. How does the ΔSST compare to modern ranges of SSTs in the area from too hot to too cold and their density/extension/calcification.

Please label the figures with the names of the corals pieces and not just have them in the caption. Figure 2, 3, and S1 in particular.

It would be good to look past the immediate analogues for the data presented in figure 3. This would allow the discerning of whether it is location, evolution or environment causing the anomalously low values from the MMCO. The modern data is definitely offset from the Miocene (both yours and the other published data), but how do the levels of preservation compare between the Miocene corals? Could it be due to 15 million years of evolution or a species level effect?

Minor points:

L18 – I'm not deep resilience is the best term to use here, it is not immediately clear what it means and to be so close to the start of the abstract it is not ideal. Spelling it out or using a more meaningful term would be nice here.

L25: I'm not sure this is strictly true from looking at the data. And I question whether it is really an adaptation or just their regular mode of life during this epoch. I cannot find other reference to "complex reefs" other than right at the end of the conclusion. I am not sure we can say with confidence if the low calcification is a product of general climate or the extremely isolated nature of the basin.

L42: I would be careful with "all growth parameters" here, they are still proxy records and coral fragments, so the data is not perfect. Perhaps nuance?

L46: Porites [corals] are.... As it is the creatures not the genus building the reef.

L57–60: Not as hot as other periods where corals have lived and thrived in the geological past.

L74+: somewhere here it should say a species or why a species cannot be identified, are they all the same?

L101 and 103: use "and" instead of "but" here, they are not unrelated parameters.

L170: reference for this statement about B/Ca. And caveat with the point arrived at L176, that growth rate is also a major control on element incorporation. Mention the importance of Raleigh fractionation.

L237: presumably local CO_2 -sw, Cen CO_2 PIP can provide a good estimate of the global change through the combination of CO_2 and DIC/Omega used, which is lower, but this may have been very much amplified in the restricted basin used here.

L253: Early Miocene at least very likely does have lower seawater pH than today (or at least than preindustrial) so it may be more of a threshold response. Annotations on figure 3 would help illustrate this point about low extension rates (but also, why are extension rates low?).

L266: What is the DIC or alkalinity in this case?

Figure4: I'm not really sure what this figure really adds? Maybe I'm missing the point, is it just that there was a long period of lower than average growth before and after the bleaching? Could the information not be added to figure 2 alongside the geochemical data?

L666: please define HDB again here.

L667: make this feature clearer on the figure, it took me ages to realise what was being reference here with the white line.

References

C.D. Standish, T.B. Chalk, M. Saeed, F. Lei, M.C. Buckingham, C. D'Angelo, J. Wiedenmann, G.L. Foster, Geochemical responses of scleractinian corals to nutrient stress, *Geochimica et Cosmochimica Acta*, Volume 351, 2023, Pages 108-124, ISSN 0016-7037, <https://doi.org/10.1016/j.gca.2023.04.011>.

Communications Earth & Environment is committed to improving transparency in authorship. As part of our efforts in this direction, we are now requesting that all authors identified as 'corresponding author' create and link their Open Researcher and Contributor Identifier (ORCID) with their account on the Manuscript Tracking System prior to acceptance. ORCID helps the scientific community achieve unambiguous attribution of all scholarly contributions. You can create and link your ORCID from the home page of the Manuscript Tracking System by clicking on 'Modify my Springer Nature account' and following the instructions in the link below. Please also inform all co-authors that they can add their ORCIDs to their accounts and that they must do so prior to acceptance.

If you experience problems in linking your ORCID, please contact the Platform Support Helpdesk.

Version 1:

Decision Letter:

Dear Dr Reuter,

Your revised manuscript titled "Coral calcification at its limits during mid-Miocene warmth" has now been seen again by the original 2 reviewers, and we include their comments at the end of this message. Generally, the Reviewers are satisfied with how you addressed their comments, though Reviewer #3 still has some serious concerns regarding the data that need a better clarification/discussion.

We are interested in the possibility of publishing your study in Communications Earth & Environment, but would like to consider your responses to these concerns and assess a revised manuscript before we make a final decision on publication.

Please ensure that the revised manuscript meets the following editorial threshold:

** Validate the values reported in your data in response to the reviewer's concern and provide evidence that these values support your conclusions.

We therefore invite you to revise and resubmit your manuscript, along with a point-by-point response that takes into account the points raised. Please highlight all changes in the manuscript text file.

Please submit your point-by-point responses as a separate file, distinct from your cover letter where you can add responses to the Editors' comments that you do not want to be made available to the reviewers. Word files are preferred. We recommend that any figures, tables or graphs that are included in the response to reviewers are also included in the main article or Supplementary Information.

Please use the following link to submit your revised manuscript, point-by-point response to the referees' comments (which should be in a separate document to any cover letter), a tracked-changes version of the manuscript (as a PDF file) and the completed checklist:

Link Redacted

We hope to receive your revised paper within six weeks; please let us know if you aren't able to submit it within this time so that we can discuss how best to proceed. If we don't hear from you, and the revision process takes significantly longer, we may close your file. In this event, we will still be happy to reconsider your paper at a later date, as long as nothing similar has been accepted for publication at Communications Earth & Environment or published elsewhere in the meantime.

Please do not hesitate to contact us if you have any questions or would like to discuss these revisions further. We look forward to seeing the revised manuscript and thank you for the opportunity to review your work.

Best regards,

Nadine Schubert, PhD
Editorial Board Member
Communications Earth & Environment
orcid.org/0000-0001-7161-7882

Carolina Ortiz Guerrero, PhD (she/her/ella)
Associate Editor
Communications Earth & Environment

EDITORIAL POLICIES AND FORMATTING

Editorial Policy: [Policy requirements](https://www.nature.com/documents/nr-editorial-policy-checklist.pdf) (Download the link to your computer as a PDF.)

- Behavioural and social science
- Ecological, evolutionary & environmental sciences
- Life sciences

<https://www.nature.com/documents/nr-reporting-summary.zip>

Furthermore, please align your manuscript with our format requirements, which are summarized on the following checklist: [Communications Earth & Environment formatting checklist](https://www.nature.com/documents/commsj-phys-style-formatting-checklist-article.pdf)

and also in our style and formatting guide [Communications Earth & Environment formatting guide](https://www.nature.com/documents/commsj-phys-style-formatting-guide-accept.pdf) .

*** DATA: Communications Earth & Environment endorses the principles of the Enabling FAIR data project (<http://www.copdess.org/enabling-fair-data-project/>). We ask authors to make the data that support their conclusions available in permanent, publically accessible data repositories. (Please contact the editor if you are unable to make your data available).

All Communications Earth & Environment manuscripts must include a section titled "Data Availability" at the end of the Methods section or main text (if no Methods). More information on this policy, is available at <http://www.nature.com/authors/policies/data/data-availability-statements-data-citations.pdf>.

If a community resource is unavailable, data can be submitted to generalist repositories such as [figshare](https://figshare.com/) or [Dryad Digital Repository](http://datadryad.org/). Please provide a unique identifier for the data (for example a DOI or a permanent URL) in the data availability statement, if possible. If the repository does not provide identifiers, we encourage authors to supply the search terms that will return the data. For data that have been obtained from publically available sources, please provide a URL and the specific data product name in the data availability statement. Data with a DOI should be further cited in the methods reference section.

REVIEWER COMMENTS:

Reviewer #2 (Remarks to the Author):

I was asked to re-review “Coral calcification during mid-Miocene warmth.” I thought that the changes improved the manuscript. I would also like to thank the author for considering and giving reasonable responses to all my queries. I am happy for the article to be published with just one minor suggestion. Good job

Line 53 remove Middle.

Reviewer #3 (Remarks to the Author):

This is my second time seeing this manuscript, and many things have been improved from the first reading, especially the discussion around SSTs. I thank the authors for their clarifications and the response to review, and hope they agree the manuscript is in a better form now. I do still however, have one or two minor issues remaining to further improve the clarity before publication.

The TE (notably B) data still seem high? What is the literature basis for values of B/Ca in Porites corals being as high as 700 $\mu\text{mol/mol}$ – this is why I referred to the influence of nutrients previously, as that can drive such high values, where the B/Ca and Sr/Ca are also well correlated (and not at all related to run-off either as they are from culture experiments in Standish et al. 2023). But also why I mentioned that the JcP values are a bit higher than I would expect (13–15%). The cycles are also the largest I've seen in the literature. This is not to say that the data are incorrect, but that the values are worth making reference to, whether they are influenced by analytical, environmental or other factors (and what the uncertainty on that might be).

L241 (old L237): What I had meant here is that changes in local CO_3sw might be quite different to changes in overall ocean CO_3 , both in absolute values and in seasonality, due to the restricted nature of the basin. Just to caveat, that it isn't necessarily (global) ocean acidification that would decrease them.

Maybe we confused each other on the point regarding Rayleigh fractionation. I wasn't saying that you had conflated growth rate and elemental incorporation, merely that, as you point out in your response to review, it might play a secondary role, in X/Ca cycles. It would be nice to add a summarised and shortened version of your response to review in the manuscript. E.g. mention why you think it is not related to Rayleigh fractionation for B.

Minor points:

L35: Start the sentence with “Assessing”.

L88: clarify the phrasing of non-framework forming level-bottom communities. Are they “bottom level” or none framework forming communities with level bottoms?

L92: sclerochronological seems unnecessary here. The word is never used again, nor defined.

L230: replace while with “and”.

L278: The previous comment asking about DIC here was regarding the seeps near Papua New Guinea. My apologies that that was ambiguous. Is the low pH potentially associated with very high DIC?

Communications Earth & Environment is committed to improving transparency in authorship. As part of our efforts in this direction, we are now requesting that all authors identified as ‘corresponding author’ create and link their Open Researcher and Contributor Identifier (ORCID) with their account on the Manuscript Tracking System prior to acceptance. ORCID helps the scientific community achieve unambiguous attribution of all scholarly contributions. You can create and link your ORCID

from the home page of the Manuscript Tracking System by clicking on 'Modify my Springer Nature account' and following the instructions in the link below. Please also inform all co-authors that they can add their ORCID to their accounts and that they must do so prior to acceptance.

Version 2:

Decision Letter:

Dear Dr Reuter,

We apologize for our delay in sending this decision letter.

Your revised manuscript titled "Coral calcification at its limits during mid-Miocene warmth" has now been seen by a new reviewer (#4), whose comments appear below. In light of their advice we are delighted to say that we are happy, in principle, to publish a suitably revised version in Communications Earth & Environment, provided you expand your discussion to clarify the interpretation of your geochemical data and address all other discussion points from the reviewer.

We therefore invite you to revise your paper one last time to address the remaining concerns of our reviewer. At the same time we ask that you edit your manuscript to comply with our format requirements and to maximise the accessibility and therefore the impact of your work.

EDITORIAL REQUESTS:

*****Please take care to match our formatting and policy requirements. We will check revised manuscript and return manuscripts that do not comply. Such requests will lead to delays. *****

SUBMISSION INFORMATION:

OPEN ACCESS:

Communications Earth & Environment is a fully open access journal. Articles are made freely accessible on publication. For further information about article processing charges, open access funding, and advice and support from Nature Research, please visit <https://www.nature.com/commsenv/open-access>

Link Redacted

Best regards,

Carolina Ortiz Guerrero, Ph.D.
Associate Editor,
Communications Earth & Environment
Consulting Editor, Communications Sustainability

Nadine Schubert, PhD
Editorial Board Member
Communications Earth & Environment
orcid.org/0000-0001-7161-7882

REVIEWERS' COMMENTS:

Reviewer #4 (Remarks to the Author):

Overall this is an interesting study of coral calcification response to warm climates in the mid-Miocene, with implications for a similar scenario under future climate change. The main approach used in this study, namely annual cycles in stable isotopes and trace elements associated with seasonal growth bands, has been known for decades, but studies that actually use them to infer the calcification physiology of corals are not so common, making this paper a relatively novel piece of work. The study is additionally strengthened by the well-preserved coral samples from the mid-Miocene warm interval that seem to span a gradual cooling event. I think the data presented are generally of good quality and worthy of consideration for publication, but would like to see some more details (as listed below) addressed to clarify some key points and make it easier to follow by readers who are not familiar with the implications of different geochemical proxies.

1. In the section “indication for temperature-induced growth stress”, although I understand that many biological factors can influence the seasonal amplitudes in $\delta^{18}\text{O}$ and Sr/Ca, I think it would be really helpful to try to estimate their respective indications for the seasonality of temperature and see if they give consistent results. The authors did it for $\delta^{18}\text{O}$, and I wonder what kind of results we get for Sr/Ca based on the range of Sr/Ca-temperature slopes for modern *Porites* corals, and which one might be consistent with estimates from $\delta^{18}\text{O}$. In that way, the authors may also be able to infer more about the physiology of the corals, as some studies have pointed out that the temperature sensitivity of Sr/Ca to temperature could be related to symbiotic effects as well as Rayleigh distillation, in which case stronger Rayleigh distillation is usually associated with lower Sr/Ca values (Cohen et al., 2002; Sinclair, 2005; Gagnon et al., 2007; Gaetani et al., 2011).

2. The authors mentioned that B/Ca in these Miocene corals are generally higher than modern *Porites* (Line 298). I think an important question is by how much and what are the implications. The Jcp-1 values measured by the authors with the LA-ICP-MS method are 13.8% higher than the accepted value, although they are barely within 2σ range. So it would be helpful to put some modern reference in the text, as well as in Figure 3. I think the same argument can be made for Sr/Ca, in which case a visual comparison with modern corals is needed. Regarding the meaning of B/Ca in coral skeletons, it is still controversial and the authors should be more explicit as to what they think the proxy indicates in their samples. For example, in Line 231, the authors say B/Ca is sensitive to the carbonate chemistry up-regulation of the cf. The term “carbonate chemistry up-regulation” is confusing because it does not say which parameter is up-regulated, pH, $[\text{CO}_3]_{\text{cf}}$, DIC or some combination of them. I expect such an up-regulation to increase B/Ca in coral skeletons, as the authors say in Line 232, but some of the references cited (e.g. DeCarlo et al., 2018) instead show that higher B/Ca mean lower $[\text{CO}_3]_{\text{cf}}$, which is used later by the authors to infer that the mid-Miocene corals may have lived in lower $[\text{CO}_3]_{\text{sw}}$ given their higher B/Ca compared to modern corals. Although the exact mechanism of the B/Ca proxy is not totally clear, I think the authors need to be more explicit about what they think their B/Ca data indicate in terms of $[\text{CO}_3]_{\text{cf}}$ and $[\text{CO}_3]_{\text{sw}}$.

3. The comparison between the early, mid- and late Miocene corals is interesting (starting from line 275). In Figure 2, it looks like early Miocene growth rates fall closer to the mid-Miocene trend, while the late Miocene rates are on the low end of the modern coral trend. Supposedly this could be related to the cooling event. Maybe this is worth some discussion.

4. For many of the correlations mentioned in the text, only correlation coefficients and p-values are provided. I would like to at least see some cross plots of correlations between the isotopes and Me/Ca ratios, as well as with skeletal density and growth rates in the supplementary material. There are a few places where I feel quite confused in terms of the correlations referred to by the authors. For example, in Line 198, $r=0.93$ between $\delta^{18}\text{O}$ and $\delta^{13}\text{C}$, I think only applies to the negative $\delta^{13}\text{C}$ excursion event instead of the whole time series, and the authors should be more explicit about that (because overall $\delta^{18}\text{O}$ and $\delta^{13}\text{C}$ are negatively correlated in these coral samples over seasonal scales, as mentioned in Line 223). Also if a kinetic fractionation mechanism is inferred, $\delta^{18}\text{O}$ - $\delta^{13}\text{C}$ cross plots should show characteristic slopes (~ 2).

Minor Edits (line numbers in the pdf file with revision marks, same above)

Line 87: characterized by “a” lack of

Line 118: there’s a “?” that needs to be replaced

Line 198: $p < 0.0001$

Line 314: in present day corals, “in which seasonal cycles in B/Ca are associated with” metabolic DIC supply

Line 315: replace “Despite” with “Although”, or “were” with “being”

Line 470: the “r” in “yr” should not be superscripted

Line 471: You mean 100.09 instead of 100.9 for the molar weight of CaCO_3 ?

Dear Editor,

On behalf of the co-authors, I am sending you the revised manuscript “Coral calcification at its limits during mid-Miocene warmth”. Below you will find our detailed responses to the reviewers’ comments.

With kind regards,

Markus Reuter

Reviewer #2:

The way the article is laid out right now makes it hard to read. Important facts are getting lost in the article (I was not aware until the end that these were subtropical corals). Also, the different coral sites are getting mixed up. I will have more specific comments below, but I highly recommend a results section where you detail what the cores showed regarding growth rates and geochemical measurements before a separate discussion section. This would make it easier to follow your reasoning.

To improve the data presentation and understanding of our reasoning, we followed the specific comments of reviewer #2 in the revision. In particular, we have added a new table (Table 1), which summarizes the characteristics of the three *Porites* sclerochronological records from the Central Paratethys. In addition, we added a new chapter containing information on the localities and stratigraphic position of the corals studied. Also, the information that the corals are from a subtropical environment is given earlier in the abstract and the new chapter “Localities and stratigraphic background” of the revised manuscript.

Abstract: You need to mention that these corals are growing in subtropical conditions here. In your last sentence, you state that the important finding is that corals were stressed in subtropical environments.

In the abstract of the revised manuscript, we explain that the Central Paratethys was a subtropical Sea.

Line 43-44: "However, the preservation of all growth parameters (bulk density, linear extension rate and calcification rate) and of the original geochemical composition is a very rare feature in fossil corals, particularly further back in the geological record." This sentence reads very awkwardly because it runs on a little. I would look over the intro paragraph again because some of the sentences could be broken up and read better.

We have reworded the introduction paragraph.

Line 52: It is becoming more common to use Miocene Climatic Optimum (MCO) instead of the MMCT because it is now thought that the MCO extended into the Early Miocene.

We have replaced "MMCO" by "MCO" in the revised manuscript.

Line 54: It showed is not correct surely it is it shows in the present tense.

We use the present tense in the revised manuscript.

Line 74-87: Shouldn't this be in the intro or methods? I would not categorize this as a result.

We think neither. The information on localities and stratigraphy is provided in a new chapter 'Localities and stratigraphic background' in the revised manuscript.

Figure 1: make the site locations brighter. I had trouble finding site 1 on the figure also, you might consider adding the nicknames (U4 etc.) to the site map and age figure above.

The sample names are added to Figure 1 and the symbols for the site locations are changed for better visibility.

General Comment: I would find it helpful if you included a table here showing the age location and maybe some major measurements for the different corals. I got lost in the article, trying to remember which coral corresponded to which period. Doing this would make it easier to find this information.

We add a summary table (Table 1 in the revised manuscript) providing information on the locations, age, timing of density banding, minimum, maximum and mean values and standard deviation of the calcification parameters, $\delta^{18}\text{O}$, $\delta^{13}\text{C}$, B/Ca and Sr/Ca.

Lines 100-105: This would be much better in the table mentioned above. Right now, it is jumping around a lot, and that makes it hard to read. It would also save you some space.

The values for bulk density, extension rates and calcification rates are transferred to the new Table 1.

Line 144: It should be the interpretation

The concluding interpretation that the oxygen isotope signatures of samples U4 and Poetz are suitable for the interpretation of SST seasonality is made in the revised manuscript after the discussion of possible control factors for all three corals.

Line 156: as stated above this needs to be said earlier

The information that the Central Paratethys harbored the highest latitude warm-water reef system of the late Cenozoic during the Badenian is moved in the new chapter "Localities and stratigraphic background"

Lines 158-216: I would consider switching the discussion of U4 before Poetz. It is a little odd to talk about the older coral second and then the MMCT before the MCO. At the very least, I would mention in both these paragraphs the age and whether these are MMCT or MCO corals because this would make it easier to follow the importance of the seasonal signals.

In the revised manuscript, we first discuss the seasonal pattern of the coral U4 and then of the coral Poetz, as this corresponds to the temporal succession from the MCO to the MMCT. This should help to make the interpretation easier to follow.

170-174: this discussion of B/Ca seems to be in the wrong place it really should be a separate paragraph before or after the discussion of the coral signals.

We have restructured the chapter accordingly.

Lines 192-194: This seems to be a really important finding, given that there are very few bleaching records during the Miocene. Right now, it is buried in the center of this VERY long paragraph. Maybe a paragraph break here would emphasize this finding better.

The text has been divided by a paragraph here.

Lines 219-220: Where do these SSTs come from? if they are assumed from the low summer growth rates above this has to be stated. However, given that the authors earlier stated that they would not reconstruct SSTs from $\delta^{18}\text{O}$ (correctly) then there is no concrete evidence that SSTs were above optimal summer growth rates. Therefore, this finding needs to be explained better or qualified.

In order to clarify our interpretation in the revision, we have discussed our results in more detail, on which the considerations on seasonal temperature growth stress are based. To support our interpretation of unfavorable temperatures for coral growth in summer and winter during the MCO, we compare the reconstructed $\delta^{18}\text{O}$ -based SST amplitude for U4 with published seasonal SST estimates for the southern and central Central Paratethys. These estimates derived from an actualistic comparison of the region's fossil echinoderm fauna with the nearest living relatives and their climatic tolerances (Kroh, 2007) and are therefore independent of the seawater composition.

Lines 222-225: See above. It is also worth noting that other processes can cause stress and bleaching. These statements need to be explained better.

We have rewritten this paragraph to better explain why we attribute the seasonal growth stress to SST changes (see also the previous comment). As we are not dealing specifically with bleaching here, we do not consider a general statement on the causes for bleaching to be necessary for the discussion at this point.

Lines 284-287: This needs to be stated in the abstract as this is the first mention of subtropical conditions.

In the revised abstract we refer to the subtropical setting.

Reviewer #3:

The geochemical data which is key to this study is not presented as the best quality as given here. Tucked away in table S1 are some differences of 14–42% (in B) away from the carbonate standards, which is quite extreme. And the uncertainty is presented in a different unit to that plotted in the main figures ($\mu\text{g/g}$ vs. mmol/mol). It seems the lab has measured high here, is that corrected for in the data (which also reads high compared to most published modern data)? Or would you propose the published values are too low?

We have analyzed three different reference materials for quality control (USGS BCR-2G, USG MACS-3, JCp-1). The basaltic glass BCR-2G is used primarily to check the reproducibility of the measurements, which requires a homogeneous material. The results for this reference material are very consistent for both B and Sr, with a relative standard deviation of approximately 6% and 1% respectively. This means that variations in the data measured on the corals that are larger than this relative standard deviation for BCR-2G are considered as an actual environmental signal.

The two carbonate pressed powder tablets (MACS-3 and JCp-1) are less homogeneous than the basaltic glass. Therefore, the larger relative standard deviation for both materials and both elements is not problematic in terms of the reproducibility of the coral measurements. The results of MACS-3 and JCp-1 serve to make aware of any problems with the measurement and calibration of carbonate samples with synthetic glasses (here NIST SRM 610). For Sr, no problem is apparent for both reference materials. The measured data agree with the reference values considering uncertainties (1 sigma of the reference value and 1 SD of the measured values). For B, it can be seen that at low B concentrations in a carbonate matrix (CaCO_3), the boron signal is massively affected by an overlay of the neighboring carbon peak (May and Wiedmeyer, 1998) (C content approx. 12 wt% compared to a B concentration of approx. 0.0008 wt% or $8 \mu\text{g/g}$ in MACS-3). At higher B contents (e.g. JCp-1 with approx. $49 \mu\text{g/g}$), the interference has a much smaller effect. The boron contents of the analyzed corals are comparable to those of JCp-1 (coral *Porites* sp.) and we therefore assume that the stated B contents and data converted into molar ratios are similarly accurate.

Since in our interpretations we only use the variation of the measured values to determine seasonality, the data used are therefore suitable. In coral trace element studies element to calcium ratios are conventionally expressed as mmol/mol . We have also listed the El/Ca data as $\mu\text{g/g}$ in the data Supplement of the revised manuscript.

Being more upfront about the uncertainty and data values would be helpful to understanding the data in more detail. E.g. does a boron/calcium of $700 \mu\text{mol/mol}$ really imply a low CO_3^{2-} or could it be something else? (e.g. Standish et al. 2023, GCA).

Probably, the reviewer here refers to the nutrient effect given that there might be influxes from land. Nutrients may influence B/Ca variability, but they appear to play a secondary role since B/Ca closely aligns with Sr/Ca seasonality. This pattern is similar to findings by D'Olivo et al. (2017), where seasonality closely follows Sr/Ca and Li/Mg rather than Ba/Ca, which in their study was strongly influenced by terrestrial runoff and nutrient variability in the GBR.

It is a shame not to see the relative temperature reconstructions made more of, the authors make a great deal out of the exceptional preservation and original aragonite of the corals, and then consider the seasonality, but do not show it anywhere.

We add reconstructions of the SST amplitudes from $\delta^{18}\text{O}$ for samples U4 and Poetz in the revised Fig. 2. For Dras, the SST amplitude from $\delta^{18}\text{O}$ is not shown as we assume interference of SST with other factors (e.g., freshwater input, upwelling or colony rotation).

The above links into a point made on L 163 where the authors talk about temperatures low enough to be unfavourable to calcification, does this imply a much greater seasonality than the modern? Only a few locations now have tropical corals which completely stop extension during cold periods. How does the ΔSST compare to modern ranges of SSTs in the area from too hot to too cold and their density/extension/calcification.

Some of the main limitations from temperature proxies applied to ancient corals is the wide range in temperature sensitivities of individual corals and species differences, which add to the uncertainties associated with Sr and Ca concentrations of ancient oceans. These limitations prohibit establishing absolute temperature values. Instead inferences in paleolatitudes and growth behavior and the characteristics of the geochemical cycles are used to make these inferences.

Please label the figures with the names of the corals pieces and not just have them in the caption. Figure 2, 3, and S1 in particular.

We labeled Figures 2 and S1 with the names of the coral samples in the revised manuscript. However, for better readability, we refer to the key to symbols in Fig. 3.

It would be good to look past the immediate analogues for the data presented in figure 3. This would allow the discerning of whether it is location, evolution or environment causing the anomalously low values from the MMCO. The modern data is definitely offset from the Miocene (both yours and the

other published data), but how do the levels of preservation compare between the Miocene corals? Could it be due to 15 million years of evolution or a species level effect?

All fossil *Porites* used in Fig. 3 were assed according to the same criteria with regard to their preservation (Brachert et al., 2020). Calcification parameters were determined for all fossil *Porites* (Early Miocene, Middle Miocene, Late Miocene) used in this study by following the same protocol and using the same laboratory standards. Therefore, we rule out differences in the preservation of mineralogy and porosity. This is explicitly pointed out in the methods chapter of the revised manuscript. The taxonomy of *Porites* is one of the most problematic among the Scleractinia (Forsman et al., 2015, 2017) (see also comment to L74+). Species are therefore usually not differentiated in *Porites*-based calcification studies including the data set of modern *Porites* used in Fig. 3. Therefore and because of large uncertainties in the identification of fossil *Porites* species (see also comment to L74+) a comparison of the calcification data at species level is not possible. Since the density of the *Porites* corals before and after the MCO does not differ from that of today's *Porites* and also the extension rates of the Miocene *Porites* are close to or within the (lower) range of the values of today's *Porites* (Fig. 3), we assume that the higher calcification rates at present-day do not reflect an evolutionary development.

Minor points:

L18 – I'm not deep resilience is the best term to use here, it is not immediately clear what it means and to be so close to the start of the abstract it is not ideal. Spelling it out or using a more meaningful term would be nice here.

The term 'deep resilience' is defined by Gold and Vermeij (2023) as the ability to maintain a consistent phenotype despite environmental change over geologic timescales. In the revised version, we use the more neutral term 'past resilience' to refer to the resilience of calcifying organisms to ocean acidification/warming the geological past.

L25: I'm not sure this is strictly true from looking at the data. And I question whether it is really an adaptation or just their regular mode of life during this epoch. I cannot find other reference to "complex reefs" other than right at the end of the conclusion. I am not sure we can say with confidence if the low calcification is a product of general climate or the extremely isolated nature of the basin.

L25: We think that the interpretation of an internal pH upregulation by the fossil is well supported by our data. The ability of modern corals to up-regulate the pH of their calcifying fluid is thought to be a key mechanism to withstand long-time exposure to reduced seawater pH conditions and is assumed to play a crucial role in coral resilience to ocean acidification (e.g., Wall et al., 2017). There is no reason to assume that this does not also apply to the Middle Miocene corals.

Adaption: We have changed the sentence so that the term “adaption” is avoided.

Complex reefs: In the revision, we discuss this point in more detail at the end of the conclusion.

General climate vs. isolated basin: The different seasonal patterns in the coral samples U4 and Dras/Poetz (sinusoidal and cusped, respectively) corresponds to global climate trend (Westerhold et al., 2020) and the retreat of mangroves from the Central Paratethys following the MCO (Harzhauser et al., 2023). In addition, the Middle Miocene density minimum of *Porites* corals (Fig. 3) coincides with the reconstructed global CO₂ maximum of the MCO (The CenCO₂PIP Consortium, 2023). Both together indicate a global influence on coral growth in the Central Paratethys. However, we cannot assess the extent to which the special environment of the Central Paratethys has amplified or attenuated the effects of climate change compared to the open ocean, as we cannot quantify the local environmental parameters from our data (This also refers to the reviewer’s comment on L237 below).

L42: I would be careful with “all growth parameters” here, they are still proxy records and coral fragments, so the data is not perfect. Perhaps nuance?

We have rephrased the sentence as follows: “ ... information on the skeletal growth parameters bulk density, linear extension rate and calcification rate ...”

L46: Porites [corals] are.... As it is the creatures not the genus building the reef.

We replaced “*Porites* is ...” by “*Porites* corals are ...” in the revised manuscript.

L57–60: Not as hot as other periods where corals have lived and thrived in the geological past.

This is probably true, but the paragraph in concern and the study in general refer to the Middle Miocene. Additionally, no fossil coral calcification data (bulk density, linear extension rate and calcification rate) exist that would allow conclusions on the calcification responses during a period that was warmer than the MCO. Because diagenesis modifies skeletal porosity and density, fossil

calcification rates are often inferred from the linear extension rate. This estimation is achieved through the analysis of “ghost structures” of the original annual density bands that can be found in fossils either fully replaced and cemented by calcite or partly dissolved (Reuter et al., 2005; Brachert et al., 2006). Accordingly, these data cannot be used to make statements about ocean acidification. An exception is a calcification dataset from *Astreopora* corals representing the Middle Eocene Climatic Optimum ca. 40 million years ago, which includes skeletal bulk density, linear extension rate and calcification rate (Brachert et al., 2022). However, neither trace element proxy data nor modern and fossil reference data on the calcification of this taxon are available to assess the calcification response to Eocene ocean warming and acidification.

L74+: somewhere here it should say a species or why a species cannot be identified, are they all the same?

The taxonomy of *Porites* is one of the most challenging among the Scleractinia. This is due to the high variability of skeletal structures within individual coralla and geographic variation across the extensive range of many nominal species. Furthermore, the incongruence between morphological and molecular systematics has left the delimitation of many species uncertain (Forsman et al., 2015, 2017). For these reasons, *Porites*-based coral (paleo-)climate and calcification studies avoid species-level differentiation, which also applies to the reference dataset of modern and fossil *Porites* used in this study (Fig. 3). Given the minimal differences among the Miocene *Porites* species in the Mediterranean region (Chevalier, 1961), we have therefore also refrained from determining the fossil *Porites* presented here to the species level.

This paragraph is added as taxonomic remarks in the methods section of the revised manuscript.

L101 and 103: use “and” instead of “but” here, they are not unrelated parameters.

“And” is used instead of “but” here in the revised manuscript.

L170: reference for this statement about B/Ca. And caveat with the point arrived at L176, that growth rate is also a major control on element incorporation. Mention the importance of Raleigh fractionation.

Holcomb et al. (2016) and DeCarlo et al. (2018) have been inserted as references for the statement that B/Ca ratio of the coral skeleton primarily depends on $[\text{CO}_3^{2-}]_{\text{cf}}$. We think the reviewer is getting confused here, we are not stating that there is a dependency of growth rate and element

incorporation, it is simply a stretching effect caused by a change in sampling resolution due to variations in growth. So far Rayleigh fractionation has not been properly assessed for B/Ca, but it is likely to play a secondary role due to the upregulation process. Here is a quote from DeCarlo et al., 2018: “While a low partition coefficient causes Rayleigh fractionation for elements in a closed system (e.g., coral $[Mg]/[Ca]_{cf}$) (Gaetani and Cohen, 2006), $[CO_3^{2-}]_{cf}$ is elevated relative to seawater and is modified by CO_2 diffusion and pH up-regulation (i.e., it is not in a closed system) (Adkins et al., 2003; Cai et al., 2016), meaning that $[B]/[CO_3^{2-}]_{cf}$ is likely not changed substantially due to skeletal aragonite precipitation. Therefore, boron-based proxies are thought to be largely dependent on carbonate chemistry alone (Trotter et al., 2011; McCulloch et al., 2017).”

L237: presumably local CO_3^{2-} _{sw}, CenCO2PIP can provide a good estimate of the global change through the combination of CO_2 and DIC/Omega used, which is lower, but this may have been very much amplified in the restricted basin used here.

We don't quite understand the point the reviewer is trying to make.

L253: Early Miocene at least very likely does have lower seawater pH than today (or at least than preindustrial) so it may be more of a threshold response. Annotations on figure 3 would help illustrate this point about low extension rates (but also, why are extension rates low?).

Extension rate: All Miocene calcification records of *Porites* originate from the northern edge of the tropical reef belt (Perrin and Bosselini, 2012), which had a wider latitudinal extent in the Miocene than it has today (Perrin, 2002). The low extension rates of Miocene *Porites* corals (Fig. 3) could therefore be an expression of the low mean annual SST and high seasonal environmental variability at high latitudes.

Seawater pH: Despite a higher atmospheric CO_2 concentration (The CenCO₂PIP Consortium) and a lower pH value of the seawater during the Early and Later Miocene compared to today (Sosdian et al., 2018), *Porites* does not show a lower density at these times (Fig. 3). This suggests that the corals were able to balance the lower pH of the seawater (Sosdian et al., 2018) by their control over pH_{cf} . Only under the higher CO_2 conditions of the MCO, *Porites* corals seem to reach their physiological limit of pH homeostasis leading to a decrease of the skeletal bulk density.

We have addressed both points in the discussion (“Effect of ocean acidity”) in the revised manuscript.

L266: What is the DIC or alkalinity in this case?

We cannot reconstruct DIC_{sw} from the coral data as we don't have the pH typically obtained from the boron isotopes. We could use the estimated sw pH value of 7.6, but this would miss the large seasonality variability, so not sure if this approximation would be meaningful at all.

Figure 4: I'm not really sure what this figure really adds? Maybe I'm missing the point, is it just that there was a long period of lower-than-average growth before and after the bleaching? Could the information not be added to figure 2 alongside the geochemical data?

Figure 2 is already quite detailed, showing the seasonal amplitude changes on the Y-axes and the growth rates shown on the X-axes. Therefore, we have decided to move Fig. 4 in the Supplement.

L666: please define HDB again here.

The abbreviation HDB is written out here as "high density band" in the revised manuscript.

L667: make this feature clearer on the figure, it took me ages to realise what was being reference here with the white line.

The white line is replaced by a yellow arrow in the revised Supplementary Fig. 1.

References

Adkins, J. F., Boyle, E. A., Curry, W. B. & Lutringer, A. Stable isotopes in deep-sea corals and a new mechanism for "vital effects". *Geochim. Cosmochim. Acta* **67**, 1129-1143 (2003).

Brachert, T. C., Reuter, M., Kroeger, K. F., Lough, J. Coral growth bands: A new and easy to use paleothermometer in paleoenvironment analysis and paleoceanography (late Miocene, Greece). *Paleoceanography* **21**, PA4217 (2006).

Brachert, T. C. et al. An assessment of reef coral calcification over the late Cenozoic. *Earth-Sci. Rev.* **204**, 103154 (2020).

Brachert, T. C., Felis, T., Gagnaison, C., Hoehle, M., Reuter, M. & Spreter, P. M. Slow-growing reef corals as climate archives: A case study of the Middle Eocene Climatic Optimum 40 Ma ago. *Sci. Adv.* **8**, eabm3875 (2022).

Cai, W.-J., Ma, Y., Hopkinson, B. M., Grottoli, A. G., Warner, M. E., Ding, Q., Hu, X., Yuan, X., Schoepf, V., Xu, H., Han, C., Melman, T. F., Hoadley, K. D., Pettay, D. T., Matsui, Y., Baumann, J. H., Levas, S., Ying, Y. & Wang, Y. Microelectrode characterization of coral daytime interior pH and carbonate chemistry. *Nat. Commun.* **7**, 11144 (2016).

Chevalier, J.-P. Recherches sur les madréporaires et les formations récifales miocènes de la Méditerranée occidentale. *Mém. Soc. Géol. France* **93**, 1–558 (1961).

DeCarlo, T. M., Holcomb, M. & McCulloch, M. T. Reviews and syntheses: Revisiting the boron systematics of aragonite and their application to coral calcification. *Biogeosci.* **15**, 2819–2834 (2018).

D'Olivo, J. P. & McCulloch, M. T. Response of coral calcification and calcifying fluid composition to thermally induced bleaching stress. *Sci. Rep.* **7**, 2207 (2017).

Fantazzini, P. et al. Gains and losses of coral skeletal porosity changes with ocean acidification acclimation. *Nat. Commun.* **6**, 7785 (2015).

Forsman, Z., Wellington, G. M., Fox, G. E. & Toonen, R. J. Clues to unraveling the coral species problem: distinguishing species from geographic variation in *Porites* across the Pacific with molecular markers and microskeletal traits. *PeerJ* **3**, e751 (2015).

Forsman, Z. H., Knapp, I. S. S., Tishammer, K., Eaton, D. A. R., Blecaid, M. & Toonen, R. J. Coral hybridization or phenotypic variation? Genomic data reveal gene flow between *Porites lobata* and *P. compressa*. *Mol. Phylogenetics Evol.* **111**, 132–148 (2017).

Gaetani, G. A. & Cohen, A. L. Element partitioning during precipitation of aragonite from seawater: A framework for understanding paleoproxies. *Geochim. Cosmochim. Acta* **70**, 4617–4634 (2006).

Gold, D. A. & Vermeij, G. J. Deep resilience: An evolutionary perspective on calcification in an age of ocean acidification. *Front. Physiol.* **14**, 1092321 (2023).

Harzhauser, M., Guzhov, A., Landau, B. M., Kern, A. K. & Neubauer, T. A. Oligocene to Pleistocene mudwhelks (Gastropoda: Potamididae, Batillariidae) of the Eurasian Paratethys Sea – Diversity, origins and mangroves. *Palaeogeogr. Palaeoclimatol. Palaeoecol.* **630**, 111811 (2023).

Holcomb, M., DeCarlo, T. M., Gaetani, G. A. & McCulloch, M. Factors affecting B/Ca ratios in synthetic aragonite. *Chem. Geol.* **437**, 67-76 (2016).

Kroh, A. Climate changes in the Early to Middle Miocene of the Central Paratethys and the origin of its echinoderm fauna. *Palaeogeogr. Palaeoclimatol. Palaeoecol.* **253**, 169-207 (2007).

May, T. W. & Wiedmeyer, R. H. A Table of polyatomic interferences in ICP-MS. *At. Spectrosc.* **19(5)**, 150-155 (1998).

McCulloch, M. T., D'Olivo, J. P., Falter, J., Holcomb, M. & Trotter, J. A. Coral calcification in a changing World and the interactive dynamics of pH and DIC upregulation. *Nat. Commun.* **8**, 15686 (2017).

Perrin, C. Tertiary: the emergence of modern reef ecosystems. In *Phanerozoic Reef Patterns* (eds Kiessling, W., Flügel, E. & Golonka, J.). 587–621 (SEPM, 2002).

Perrin, C. & Bosellini, F. R. Paleobiogeography of scleractinian reef corals: Changing patterns during the Oligocene–Miocene climatic transition in the Mediterranean. *Earth-Sci. Rev.* **111**, 1–24 (2012).

Reuter, M., Brachert, T. C. & Kroeger, K. F. Diagenesis of growth bands in fossil scleractinian corals: identification and modes of preservation. *Facies* **51**, 146-159 (2005).

Sosdian, S. M., Greenop, R., Hain, M. P., Foster, G. L., Pearson, P. N. & Lear, C. H. Constraining the evolution of Neogene ocean carbonate chemistry using the boron isotope pH proxy. *Earth Planet. Sci. Lett.* **498**, 362–376 (2018).

The CenCO₂PIP Consortium. Toward a Cenozoic history of atmospheric CO₂. *Science* **382**, eadi5177 (2023).

Trotter, J., Montagna, P., McCulloch, M., Silenzi, S., Reynaud, S., Mortimer, G., Martin, S., Ferrier-Pagès, C., Gattuso, J. P. & Rodolfo-Metalpa, R. Quantifying the pH “vital effect” in the temperate zooxanthellate coral *Cladocora caespitosa*: Validation of the boron seawater pH proxy. *Earth Planet. Sci. Lett.* **303**, 163–173 (2011).

Westerhold, T. et al. An astronomically dated record of Earth’s climate and its predictability over the last 66 million years. *Science* **369**, 1383–1387 (2020).

Reviewer #2

Line 53 remove Middle.

The word “Middle” was deleted from the revised manuscript.

Reviewer #3

The TE (notably B) data still seem high? What is the literature basis for values of B/Ca in Porites corals being as high as 700 μmol/mol – this is why I referred to the influence of nutrients previously, as that can drive such high values, where the B/Ca and Sr/Ca are also well correlated (and not at all related to run-off either as they are from culture experiments in Standish et al. 2023). But also why I mentioned that the JcP values are a bit higher than I would expect (13–15%). The cycles are also the largest I’ve seen in the literature. This is not to say that the data are incorrect, but that the values are worth making reference to, whether they are influenced by analytical, environmental or other factors (and what the uncertainty on that might be).

The systematic alignment of the geochemical proxy variations with the skeletal density banding patterns (Fig. 3) indicates a preservation of primary geochemical signals in all three *Porites* fossils and the annual nature of the recorded cycles. Nevertheless, trace element concentrations, particularly B, appear elevated in the fossil corals compared to modern *Porites* (D’Olivo et al., 2018; Canesi et al., 2023). To evaluate the quality of measurement and calibration, we compare the JcP-1 values measured in this study with mean values for JcP-1 from the GeoReM database (<http://georem.mpch-mainz.gwdg.de/>, Application Version 27) based on literature data. Published values (Sr: n = 8; B: n = 6) range from $6670 \pm 230 \mu\text{g/g}$ to $7500 \pm ? \mu\text{g/g}$ for Sr and from $47.7 \pm 1.2 \mu\text{g/g}$ to $52.4 \pm 2.2 \mu\text{g/g}$ for B. Our JcP-1 values (Supplementary Tab. 1) are consistent with these ranges. Therefore, absolute values for Middle Miocene corals are inferred to reflect internal or external factors influencing trace element incorporation in the coral skeletons. When compared to modern corals, the fossil coral calcification and trace element values are closest to those from extreme *Porites* environments, such as the Galápagos (Manzello et al., 2018; Thompson et al., 2022). We include this paragraph in the chapter “Preservation, skeletal growth characteristics and geochemical patterns”.

Maybe we confused each other on the point regarding Rayleigh fractionation. I wasn’t saying that you had conflated growth rate and elemental incorporation, merely that, as you point out in your response to review, it might play a secondary role, in X/Ca cycles. It would be nice to add a summarised and shortened version of your response to review in the manuscript. E.g. mention why you think it is not related to Rayleigh fractionation for B.

Sensitivity of the Sr/Ca ratios in the Middle Miocene coral skeletons may have been influenced by Rayleigh fractionation as a result of low rates of skeletal aragonite precipitation (Galochkina et al., 2023; Ram & Erez, 2023), consistent with their low calcification rates. We include this sentence in the chapter “Indication for temperature-induced growth stress” in our discussion on the uncertainties in the use of Sr/Ca as a paleo-thermometer.

In contrast to Sr/Ca, B/Ca appears to be unaffected by Rayleigh fractionation and determined solely by the carbonate chemistry of the calcifying fluid (McCulloch et al., 2017; DeCarlo et al., 2018).

Therefore, the elevated B/Ca ratios of Middle Miocene *Porites* compared to modern *Porites* (D’Olivo et al., 2018; Canesi et al., 2023) and of U4 and Dras compared to Poetz (Tab. 1) are consistent with reduced $[\text{CO}_3^{2-}]_{\text{sw}}$ in the Central Paratethys during the MCO, in accordance with the open ocean trends of surface water pH and Ω_{ar} in the late Cenozoic (Sosdian et al., 2018). However, clearly determining external influences on the B/Ca composition of Miocene coral skeletons is complicated by uncertainties in seawater elemental concentrations, carbonate chemistry, and nutrient levels on both global and local scales. We add this paragraph to the discussion on local vs. global changes in $[\text{CO}_3^{2-}]_{\text{sw}}$ in chapter “Effect of ocean acidity”.

L241 (old L237): What I had meant here is that changes in local CO_3 sw might be quite different to changes in overall ocean CO_3 , both in absolute values and in seasonality, due to the restricted nature of the basin. Just to caveat, that it isn’t necessarily (global) ocean acidification that would decrease them.

In the revised manuscript (chapter “Effect of ocean acidity”), we refer to the possibility that changes in local $[\text{CO}_3^{2-}]_{\text{sw}}$ may have differed from the changes in overall ocean $[\text{CO}_3^{2-}]$, both in absolute values and seasonality, due to the restricted nature of the Central Paratethys basin.

L35: Start the sentence with “Assessing”.

The sentence has been reworded accordingly.

L88: clarify the phrasing of non-framework forming level-bottom communities. Are they “bottom level” or none framework forming communities with level bottoms?

The sentence has been rephrased as follows: “This marginal reef system was characterized by the lack of extensive coral reefs and the dominance of low-diversity coral carpets and non-framework coral communities.”

L92: sclerochronological seems unnecessary here. The word is never used again, nor defined.

We have changed the title of the chapter to “Preservation, skeletal growth characteristics and geochemical patterns” to avoid the term “sclerochronological”. For the same reason we changed the heading of Table 1.

L230: replace while with “and”.

“While” has been replaced with “and” in the revision.

L278: The previous comment asking about DIC here (“What is the DIC or alkalinity in this case?”) was regarding the seeps near Papua New Guinea. My apologies that that was ambiguous. Is the low pH potentially associated with very high DIC?

In the revision, we provide the DIC_{sw} (2235 $\mu\text{mol kg}^{-1}$) for the low-pH_{sw} (7.4) site as given in Wall et al. (2016).

References

- DeCarlo, T. M., Holcomb, M. & McCulloch, M. T. Reviews and syntheses: Revisiting the boron systematics of aragonite and their application to coral calcification. *Biogeosci.* **15**, 2819-2834 (2018).
- D'Olivo, J. P., Sinclair, D. J., Rankenburg, K. & McCulloch, M. T. A universal multi-trace element calibration for reconstructing sea surface temperatures from long-lived *Porites* corals: Removing 'vital-effects'. *Geochim. Cosmochim. Acta* **239**, 109–135 (2018).
- Galochkina, M., Cohen, A. L., Oppo, D. W., Mollica, N. & Horton, F. Coral Sr-U thermometry tracks ocean temperature and reconciles Sr/Ca discrepancies caused by Rayleigh fractionation. *Paleoceanogr. Paleoclimatol.* **38**, e2022PA004541 (2023).
- Manzello, D. P. et al. Galápagos coral reef persistence after ENSO warming across an acidification gradient. *Geophys. Res. Lett.* **41**, 9001–9008 (2014).
- McCulloch, M. T., D'Olivo, J. P., Falter, J., Holcomb, M. & Trotter, J. A. Coral calcification in a changing World and the interactive dynamics of pH and DIC upregulation. *Nat. Commun.* **8**, 15686 (2017).
- Ram, S. & Erez, J. Anion elements incorporation into corals skeletons: Experimental approach for biomineralization and paleo-proxies. *Proc. Natl. Acad. Sci. U. S. A.* **120**, e2306627120 (2023).
- Sosdian, S. M., Greenop, R., Hain, M. P., Foster, G. L., Pearson, P. N. & Lear, C. H. Constraining the evolution of Neogene ocean carbonate chemistry using the boron isotope pH proxy. *Earth Planet. Sci. Lett.* **498**, 362–376 (2018).
- Thompson, D. et al. Marginal reefs under stress: physiological limits render Galápagos corals susceptible to ocean acidification and thermal stress. *AGU Advances* **3**, e2021AV000509 (2022).
- Wall, M. et al. Internal pH regulation facilitates *in situ* long-term acclimation of massive corals to end-of-century carbon dioxide conditions. *Sci. Rep.* **6**, 30688 (2016).

Reviewer #4:

- 1. In the section “indication for temperature-induced growth stress”, although I understand that many biological factors can influence the seasonal amplitudes in $\delta^{18}\text{O}$ and Sr/Ca, I think it would be really helpful to try to estimate their respective indications for the seasonality of temperature and see if they give consistent results. The authors did it for $\delta^{18}\text{O}$, and I wonder what kind of results we get for Sr/Ca based on the range of Sr/Ca-temperature slopes for modern *Porites* corals, and which one might be consistent with estimates from $\delta^{18}\text{O}$. In that way, the authors may also be able to infer more about the physiology of the corals, as some studies have pointed out that the temperature sensitivity of Sr/Ca to temperature could be related to symbiotic effects as well as Rayleigh distillation, in which case stronger Rayleigh distillation is usually associated with lower Sr/Ca values (Cohen et al., 2002; Sinclair, 2005; Gagnon et al., 2007; Gaetani et al., 2011).

As the reviewer correctly indicates the Sr/Ca is impacted by other processes that we cannot fully infer here, which add to the uncertainty in the seawater concentration in Sr and Ca, this is why, as indicated in the manuscript, we opted to avoid reconstructing SST from the Sr/Ca data.

However, we tested the slopes of modern Sr/Ca-SST relationships to the fossil *Porites* corals, the reconstructed annual SST amplitude is substantially higher than that estimated from coral $\delta^{18}\text{O}$. As we have at least three factors influencing the Sr/Ca (temperature, seawater concentration and Rayleigh fractionation) we opted for a conservative approach of only reporting the $\delta^{18}\text{O}$.

However, following the suggestions of the reviewer we now include an assessment of the apparent sensitivity of the Sr/Ca to SST based on the estimates from the $\delta^{18}\text{O}$ data. The SST range inferred from coral $\delta^{18}\text{O}$ requires a Sr/Ca-SST slope of about -0.1 in all three fossil *Porites*, which is about twice the sensitivity of modern *Porites* to temperature (average -0.06). In addition to temperature, Sr/Ca appears to be controlled by Rayleigh fractionation due to variations in the amount of aragonite precipitation that is influenced mainly by calcifying fluid carbonate ion concentration, which is subject to seasonal variability as well. Part of the apparent Sr/Ca temperature sensitivity of the fossil *Porites* corals could therefore result from variations in Rayleigh fractionation (influenced by rate of precipitation and renewal rate) parameters that we cannot quantify, thus limiting the application of the Sr/Ca paleothermometer. We have added this discussion to the text.

- 2. The authors mentioned that B/Ca in these Miocene corals are generally higher than modern *Porites* (Line 298). I think an important question is by how much and what are the implications. The Jcp-1 values measured by the authors with the LA-ICP-MS method are 13.8% higher than the accepted value, although they are barely within 2σ range. So it would be helpful to put some modern reference in the text, as well as in Figure 3. I think the same argument can be made for Sr/Ca, in which case a visual comparison with modern corals is needed. Regarding the meaning of B/Ca in coral skeletons, it is still controversial and the authors should be more explicit as to what they think the proxy indicates in their samples. For example, in Line 231, the authors say B/Ca is sensitive to the carbonate chemistry up-regulation of the cf. The term “carbonate chemistry up-regulation” is confusing because it does not say which parameter is up-regulated, pH, $[\text{CO}_3]_{\text{cf}}$, DIC or some combination of them. I expect such an up-regulation to increase B/Ca in coral skeletons, as the authors say in Line 232, but some of the references cited (e.g. DeCarlo et al., 2018) instead show that higher B/Ca mean lower $[\text{CO}_3]_{\text{cf}}$, which is used later by the authors to infer that the mid-Miocene corals may have lived in lower $[\text{CO}_3]_{\text{sw}}$ given their higher B/Ca compared to modern corals. Although the exact mechanism of the B/Ca proxy is not totally clear, I think the authors need to be more explicit about what they think their B/Ca data indicate in terms of $[\text{CO}_3]_{\text{cf}}$ and $[\text{CO}_3]_{\text{sw}}$.

We have added references to modern coral Sr/Ca and B/Ca data and created a new figure (Supplementary Fig. 2) to show the quantitative differences between the fossil and modern *Porites*. We have not shown this additional information graphically in Fig. 3, as the many details would overwhelm the figure. In order not to overload Fig. 3, we therefore have made a visual comparison of B/Ca and Sr/Ca in modern and the middle Miocene *Porites* corals in a separate Supplementary Figure 2.

B/Ca reflects the carbonate ion concentration ($[\text{CO}_3^{2-}]$) at the site of calcification (calcifying fluid, cf), which is primarily biologically regulated but modulated by external environmental factors. On a seasonal scale boron is lower in summer and higher in winter, which is interpreted as higher $[\text{CO}_3^{2-}]_{\text{cf}}$ in summer and lower in winter. This is also consistent with observations in modern corals and is caused by fluctuations in both pH and the DIC pool. At higher pH, the proportion of boron present as borate ion ($\text{B}(\text{OH})_4^-$) increases, and since only borate is incorporated into the carbonate lattice, elevated pH generally leads to greater boron uptake, but will be modulated by the changes in $[\text{CO}_3^{2-}]_{\text{cf}}$ which depends on the DIC and pH. We have reworded the text to clarify the role of the carbonate ion concentration in seawater and in the calcifying fluid in terms of the B/Ca concentration in coral skeleton.

3. The comparison between the early, mid- and late Miocene corals is interesting (starting from line 275). In Figure 2, it looks like early Miocene growth rates fall closer to the mid-Miocene trend, while the late Miocene rates are on the low end of the modern coral trend. Supposedly this could be related to the cooling event. Maybe this is worth some discussion.

In the revised manuscript we refer to the relationships of calcification rates between the fossil and modern *Porites* corals and the similarity with the long-term global climate trend.

- 4. For many of the correlations mentioned in the text, only correlation coefficients and p-values are provided. I would like to at least see some cross plots of correlations between the isotopes and Me/Ca ratios, as well as with skeletal density and growth rates in the supplementary material. There are a few places where I feel quite confused in terms of the correlations referred to by the authors. For example, in Line 198, $r=0.93$ between $\delta^{18}\text{O}$ and $\delta^{13}\text{C}$, I think only applies to the negative $\delta^{13}\text{C}$ excursion event instead of the whole time series, and the authors should be more explicit about that (because overall $\delta^{18}\text{O}$ and $\delta^{13}\text{C}$ are negatively correlated in these coral samples over seasonal scales, as mentioned in Line 223). Also if a kinetic fractionation mechanism is inferred, $\delta^{18}\text{O}$ - $\delta^{13}\text{C}$ cross plots should show characteristic slopes (~ 2).

The supplementary material was complemented by cross-plots illustrating the correlations mentioned in the text (Supplementary Figs. 3, 4). To avoid confusion in terms of the correlation between $\delta^{18}\text{O}$ and $\delta^{13}\text{C}$ in coral U4, we add information that the strong positive correlation refers to the first three years after the assumed high temperature stress event, when the coral grew most slowly according to the El/Ca chronologies. In addition, we show separate $\delta^{13}\text{C}$ - $\delta^{18}\text{O}$ correlations for this coral in Supplementary Fig. 4 for (a) the complete data set of stable isotopes, (b) the undisturbed $\delta^{18}\text{O}$ cycles, and (c) the interval after the disturbance event, which is characterized by a loss of $\delta^{18}\text{O}$ cyclicity.

Minor Edits (line numbers in the pdf file with revision marks, same above)

- Line 87: characterized by “a” lack of
“the” is replaced by “a”
- Line 118: there’s a “?” that needs to be replaced

The question mark is intended to indicate that the standard deviation for this value was not specified in the reference.

- *Line 198: $p < 0.0001$*

The p-value is deleted from the text and reference is made to Supplementary Fig. 4c instead.

- *Line 314: in present day corals, “in which seasonal cycles in B/Ca are associated with” metabolic DIC supply*

The sentence has been changed according to the reviewer’s suggestion.

- *Line 315: replace “Despite” with “Although”, or “were” with “being”*

“Were” is replaced with “being”.

- *Line 470: the “r” in “yr” should not be superscripted*

The formatting has been changed accordingly.

- *Line 471: You mean 100.09 instead of 100.9 for the molar weight of CaCO_3 ?*

The reviewer is correct, the molar weight of CaCO_3 has been changed to 100.09 g per mole.